# SUBSPACE-BOOSTED MODEL MERGING

## ABSTRACT

Model merging enables the combination of multiple specialized expert models into a single model capable of performing multiple tasks. However, the benefits of merging an increasing amount of specialized experts generally lead to diminishing returns and reduced overall performance gains. In this work, we empirically and theoretically analyze this limitation, proving that for Task Arithmetic-based methods, as more experts are merged, the common information dominates the task-specific information, leading to inevitable *rank collapse*. To mitigate this issue, we introduce *Subspace Boosting*, which operates on the singular value decomposed task vector space and maintains task vector ranks. *Subspace Boosting* raises merging efficacy for up to 20 experts by large margins of more than 10% when evaluated on both vision and language benchmarks. Moreover, we propose employing Higher-Order Generalized Singular Value Decomposition to quantify task similarity, offering *a new interpretable perspective on model merging*.

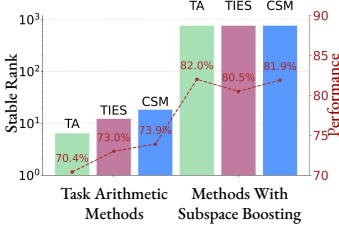
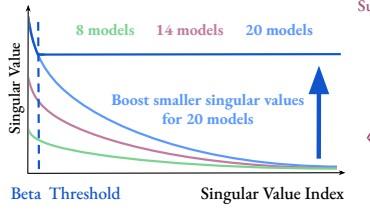
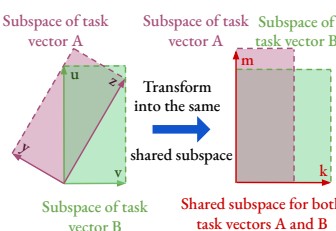

(a) The stable rank and performance for different methods before and after *Subspace Boosting*.

(b) *Subspace Boosting* boosts smaller singular values to increase the effective rank of the model.

(c) Higher-Order Generalized SVD transforms independent task vector subspaces into a shared subspace.

Figure 1: Overview of our contributions. (a) Popular merging methods such as Task Arithmetic (TA) (Ilharco et al., 2023b), TIES (Yadav et al., 2023) and Consensus Merging (CSM) (Wang et al., 2024c), suffer from **rank collapse**, correlating with low performance. (b) To prevent **rank collapse**, we introduce *Subspace Boosting*, which mitigates it by boosting neglected singular values, vastly improving performance. (c) Finally, for interpretability, we use HO-GSVD, transforming individual models to share the same subspace, enabling direct comparison.

## 1 INTRODUCTION

Training models at "foundational" scale (Bommasani et al., 2021) has significantly driven progress in the development of general-purpose models across domains, ranging from computer vision to natural language processing (Radford et al., 2021b; 2019; Brown et al., 2020; Devlin et al., 2018; Rombach et al., 2022). Despite their broad use in various downstream tasks, these foundation models still require fine-tuning to effectively adapt to specialized *expert* domains (Roth et al., 2024; Mukhoti et al., 2024), such as particular reasoning, language, or image generation. In addition, deploying and storing a growing number of *expert models* becomes unsustainable (Yadav et al., 2024b;a).

To address these problems, recent advances in model merging (Wortsman et al., 2022b; Yadav et al., 2024b; Dziadzio et al., 2025; Wang et al., 2024a) have shown significant promise. Model merg-

ing enables the creation of a ***single model*** from multiple experts (Rofin et al., 2022; Yadav et al., 2024b; 2023), while preserving the foundational generalization capability. This enables improved generalization capabilities across numerous domains, substantially simplified model inference, and the development of decentralized models.

Current approaches in model merging commonly revolve around simple scaled or filtered weight interpolation of experts. The resulting merged models outperform the base model, but, as expected, perform worse than individual task experts on their specific task. However, recent studies (Yadav et al., 2024b; Dziadzio et al., 2025) suggest that the significance of specific merging techniques may be overestimated, as most merging methods often yield comparable performance, especially at scale, indicating a gap in understanding of the underlying merging process and the weight space structure.

To bridge this gap, we first investigate the merged weight space to understand why popular merging techniques yield suboptimal performance. In particular, we investigate the task vectors, since they contain the essential task information. Our results, visualized in Fig. 1a, reveal that the merged task vectors suffer from ***rank collapse***, in which a vast majority of information is captured by the most important singular values and vectors, shown in Fig. 1b. We provide a theoretical explanation for this phenomenon: as the number of merged models increases, the common information shared across tasks accumulates, overwhelming the orthogonal, task-specific information. Consequently, the unique expert knowledge is effectively suppressed, leading the merged model to operate on a constrained subspace. This collapse can also be quantified by the *stable rank* (Zhou et al., 2010; Shukla et al., 2024; Sanyal et al., 2020) (which evaluates the "effective" rank of the matrix by disregarding small singular values). Our analysis also reveals that *rank collapse* consistently affects existing methods, seen in Fig. 1a, and that the entire model's task vector space suffers from *rank collapse*, leading to the merged task vectors operating on a constrained subspace.

To address this, we propose *Subspace Boosting*, a method that directly mitigates *rank collapse* by decomposing task vectors via Singular Value Decomposition (SVD), and explicitly boosting underutilized dimensions (Fig. 1b). Based on our theoretical insight that the task-specific signal is suppressed rather than destroyed, *Subspace Boosting* explicitly recovers these directions by boosting underutilized singular values (Fig. 1b). Therefore, by restoring the influence of these subspaces, our method significantly enhances the model's capability and performance, as shown in Fig. 1a. *Subspace Boosting* shows strong improvements across both vision and language tasks for up to 20 vision transformer experts (Dosovitskiy et al., 2021) and 8 T5 language experts (Raffel et al., 2020) of varying sizes when applied to recent merging techniques (Task Arithmetic (Ilharco et al., 2023b), TIES (Yadav et al., 2023) and Consensus Merging (Wang et al., 2024c)). For both domains, the baseline method's performance is increased by over 10%.

Finally, we use Higher-Order Generalized Singular Value Decomposition (HO-GSVD) to introduce an interpretable model merging variant (Fig. 1c). This approach decomposes task vectors into a shared space containing ***common*** and ***unique*** subspaces. This allows the comparison of task similarity or selection of optimal models via the *Alignment Matrix*, derived from the shared vector space.

Our contributions can be summarized as follows.

- We identify weight-space *rank collapse* as a crucial limitation in Task Arithmetic-based methods, which reduces generalization of the merged model. We prove that this phenomenon arises because the common information dominates the task-specific information.

- We introduce *Subspace Boosting*, a general method that mitigates *rank collapse*, which is compatible with several merging techniques, significantly improving merging efficacy across standard model merging vision and language benchmarks.

- Finally, we propose a novel framework using Higher-Order Generalized SVD on task vectors, allowing the *shared* subspace to be identified, interpreted, or used for expert selection.

## 2  RELATED WORK

Model merging (Yadav et al., 2024a; Yang et al., 2024a) has emerged as an important technique to improve post-training capabilities, and as a general toolkit to combine knowledge across different expert models (Wortsman et al., 2022a; Rame et al., 2023; Sanyal et al., 2024; Sung et al., 2023; Pari et al., 2024; Nylund et al., 2023; Zaman et al., 2023; Stoica et al., 2024; Wang et al., 2024c; He

et al., 2024; Oh et al., 2024; Shen et al., 2024; Sharma et al., 2024; Tam et al., 2024b; Goddard et al., 2024; Xiong et al., 2024; Yang et al., 2024b; Lu et al., 2024; Zheng & Wang, 2024; Nasery et al., 2024; Rofin et al., 2022; Yadav et al., 2023; Jin et al., 2023; Deep et al., 2024; Marczak et al., 2024), even across time (Roth et al., 2024; Dziadzio et al., 2025). Many model merging techniques leverage the principle of linear mode connectivity, which implies that model weights across separate training runs can be (linearly) interpolated, especially when finetuned from the same base model (Izmailov et al., 2018; Ramé et al., 2024; Neyshabur et al., 2020; Frankle et al., 2020; Ainsworth et al., 2023; Garipov et al., 2018; Entezari et al., 2022). Initial studies mainly focus on simple linear weight interpolation (Wortsman et al., 2022b; Rofin et al., 2022) or spherical linear interpolation (SLERP, (Shoemake, 1985; Ramé et al., 2024)) without particular differentiation between individual weights.

**Task Arithmetic.** Ilharco et al. (2023b) provided a task arithmetic perspective on the interpolation problem, defining finetune-to-base-weight differentials as *task vectors*. Building upon the work of Ilharco et al. (2023b), methods such as Tangent Task Arithmetic ((Ortiz-Jimenez et al., 2023), finetuning on the weight tangent space), TIES ((Yadav et al., 2023), removing low magnitude task vector entries and magnitude-based sign assignment), DARE ((Davari & Belilovsky, 2025), random weight masking over task vectors), Model Stock ((Jang et al., 2024), determining a suitable center of mass across multiple task vectors) or Breadcrumbs ((Davari & Belilovsky, 2025), cutting tail-ended task weights based on the distribution of magnitudes) have been introduced.

**Adaptive Methods.** An orthogonal line of research explores adaptive methods to find the optimal merging parameters (Lee et al., 2025; Yang et al., 2024c). By contrast, we propose a training-free method designed to improve the capabilities of existing model merging techniques. Therefore, we consider adaptive methods (Lee et al., 2025; Yang et al., 2024c) as distantly-related.

**SVD-Based.** Similar to our research, several existing works employ Singular Value Decomposition (SVD) in the context of model merging (Stoica et al., 2024; Choi et al., 2025; Marczak et al., 2025; Gargiulo et al., 2025). For example, Gargiulo et al. (2025) propose TSV-Merge, which reduces task interference by compressing layer-wise task matrices to their essential singular vectors and then decorrelating them. Similarly, Marczak et al. (2025) introduced Iso-C and Iso-CTS, two model merging methods that enhanced the performance of model merging by introducing task-specific subspaces. While these methods leverage SVD, our work is the first to diagnose, quantify and **prove** the phenomenon of *rank collapse* for Task Arithmetic-based methods. By mitigating *rank collapse*, our method substantially improves Task Arithmetic-based methods while being over 6x more time-efficient than the above SVD-based methods. Finally, we are the first to leverage HO-GSVD for merging, introducing a novel framework for interpreting task similarity and enabling principled expert selection.

# 3 RANK COLLAPSE IN MODEL MERGING

In this section, we explore the prevalent phenomenon of ***rank collapse*** in the weight space of merged models. We begin by introducing the following notation and background.

**Background.** Given a set of $N$ expert models $\{M_1, M_2, \ldots, M_N\}$ finetuned for different tasks from the same pretrained model $M_{base}$, the goal of model merging is to obtain the model $M_m$ that is capable of solving all associated expert tasks. Let $\theta_{base}$ be the parameters of $M_{base}$, and $\theta_i$ be the parameters of each expert $M_i$, $1 \leq i \leq N$. Following Ilharco et al. (2023a), we define the ***task vectors*** as the weight differential $\Delta_i = \theta_i - \theta_{base}$, $1 \leq i \leq$. Using this notation, the merged parameters $\theta_m$, defined as task arithmetic through linear interpolation, are expressed as: $\theta_m = \theta_{base} + \alpha \sum_i^N \Delta_i$, with $\alpha$ representing the scalar merging coefficient. We also define $\Delta_m = \Delta_1 + \cdots + \Delta_N$ as the total task vector.

**Rank Collapse During Merging.** To determine why model merging performance degrades as more models are merged, we investigate the subspaces spanned by the task vectors, as they isolate the knowledge specific to each task. In particular, we investigate whether merged task vectors suffer from rank collapse. We measure the rank collapse with the *stable rank* and *cumulative energy rank* (in Supplementary Sec. D). The *Stable rank* uses SVD, and for a matrix $A \in \mathbb{R}^{m \times n}$, is defined as:

$$A = U\Sigma V^T, \tag{1}$$

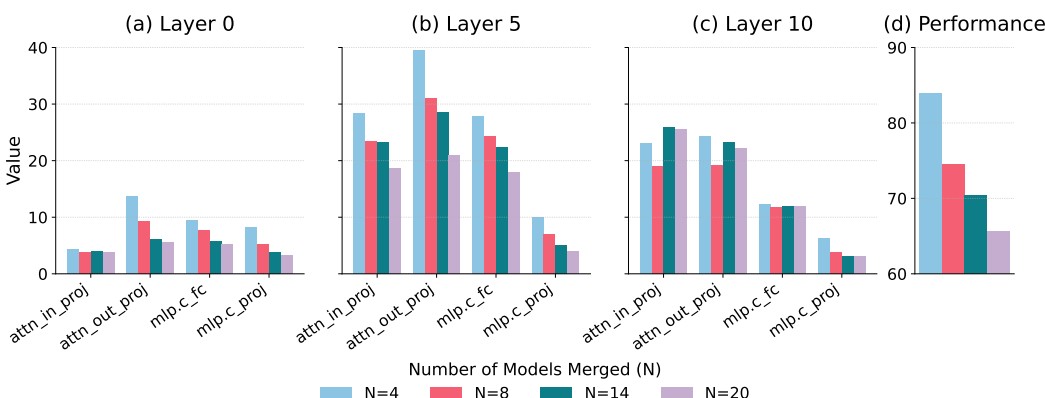

Figure 2: Stable rank in merged ViT-B/16 models. (a-c) The stable rank is decomposed across various attention and MLP sublayers of three layer blocks. (d) As more models are merged, the stable rank decreases across a majority of layers, strongly correlating with the performance.

with the orthogonal matrix $U \in \mathbb{R}^{m \times m}$, also denoted as *left* singular vectors, $\Sigma \in \mathbb{R}^{m \times n}$ the diagonal matrix containing singular values $\sigma_1 \geq \sigma_2 \geq \cdots \geq \sigma_n \geq 0$, and the *right* singular vectors $V \in \mathbb{R}^{n \times n}$.

Using this decomposition, the *stable rank* $\mathcal{M}_{\text{stable}}$ denotes the "effective rank" of the weight matrices, similar to computing the rank of the matrix by ignoring the smaller singular values:

$$\mathcal{M}_{\text{stable}} = \frac{\sum_i \sigma_i^2}{\max_i \sigma_i^2} = \frac{\|A\|_F^2}{\|A\|_2^2}. \tag{2}$$

A small stable rank indicates that most of the information is concentrated in only a few dimensions. This implies that the weight projections operate within a limited subspace, under-utilizing the full vector space.

In Fig. 2, we investigate the stable rank of merged ViT-B/16 models with Task Arithmetic. As more models are merged, we clearly observe that the stable rank decreases for a large majority of sublayers. While the problem grows in complexity as the model contains more tasks, the subspace actually contracts, contradicting the expectation that a larger set of tasks would require a higher-rank representation. This establishes a strong correlation between rank collapse and model performance degradation. Similarly, in Fig. 1a, once the stable rank increases, the performance also drastically improves across the baselines: Task Arithmetic (Ilharco et al., 2023b) (in green), TIES (Yadav et al., 2023) (in pink) or Consensus Merging (CSM) (Wang et al., 2024b) (in blue).

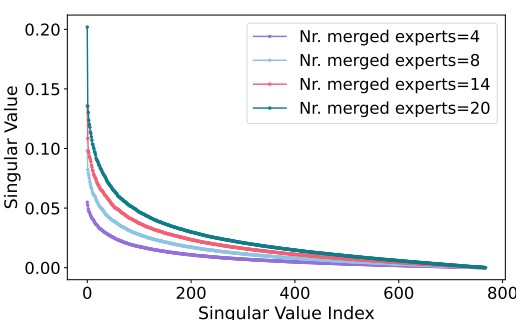

Figure 3: **Evolution of the Singular Value Distribution.** As more experts are merged, higher absolute and relative mass is placed on fewer singular vectors; encouraging the rank collapse. This indicates that information becomes concentrated in fewer dominant dimensions.

Fig. 3 further illustrates rank collapse by visualizing the singular value distributions for the models. The ordering is clear: as more models are merged, the largest singular values increase more drastically compared to the smallest singular values. For example, the largest singular value for 20 merged models ($\sigma \approx 0.20$) is $4\times$ higher than for 4 merged models ($\sigma \approx 0.05$). This also highlights that the largest singular values scale linearly, which is expected assuming they correspond to the common information. For theoretical results supporting this observation, refer to Prop. 3. For more empirical results across a number of merged models and merging methods, please see Supp. Sec. D.

## 3.1 THEORETICAL ANALYSIS OF RANK COLLAPSE

Following our empirical observations of rank collapse, we now formalize this behavior by modeling task vectors as a combination of common and task-specific information. The following propositions establish that **rank collapse is an inherent property of Task Arithmetic merging**.

**Proposition 1** (Singular Value Decay of Averaged Task-Specific Information). *Let $\Delta_m$ be the average of $N$ task vectors, decomposed into common and task-specific (noise) components. As $N \to \infty$: The singular values of the task-specific subspace decay at a rate of $\mathcal{O}(1/\sqrt{N})$ compared to the singular values of the common subspace, which remain asymptotically constant at $\mathcal{O}(1)$.*

*Proof.* *See Appendix E.1 for the full derivation relying on the pairwise orthogonality of task-specific information (Ilharco et al., 2023b).*

Consequently, the gap between the common subspace and task-specific subspace widens. In other words, as $N \to \infty$, **when averaging task vectors, the task-specific singular values decay to 0**, hence, any non-zero singular values correspond to common components.

**Proposition 2** (Asymptotic Stable Rank Collapse). *Let $\mathcal{M}_{stable} = \|A\|_F^2 / \|A\|_2^2$ denote the stable rank of a matrix. Consider the averaged task vector $\Delta_m$, decomposed into a shared common component $\Delta_{common}$ and a task-specific component $\Delta_{unique}$.*

*Under the condition that the task-specific energy decays asymptotically as $\|\Delta_{unique}\|_F \leq \mathcal{O}(1/\sqrt{N})$ from Proposition 1, the stable rank of the merged model collapses to the stable rank of the common subspace as more task vectors are merged:*

$$\lim_{N \to \infty} \mathcal{M}_{stable}(\Delta_m) = \mathcal{M}_{stable}(\Delta_{common}). \tag{3}$$

*Proof.* *For the full proof, refer to Proposition 2 in the Appendix.*

Thus, the stable rank effectively captures the impact of averaging an increasing number of task vectors. **As more task vectors are merged, the effective rank of the model collapses to that of the common components.** This is also empirically validated in Fig. 2. As more models are merged, the corresponding stable rank decreases.

**Proposition 3** (Inherent Limitation of Task Arithmetic-Based Merging). *Let the merged task vector be defined by standard Task Arithmetic as $\Delta_m = \alpha \sum_{i=1}^{N} \Delta_i$ for any scalar merging coefficient $\alpha \in \mathbb{R}$. Let $\Delta_m = \Delta_{common} + \Delta_{unique}$. As $N \to \infty$, the ratio of the spectral magnitude of the common subspace to the task-specific subspace diverges:*

$$\frac{\|\Delta_{common}\|_2}{\|\Delta_{unique}\|_2} \geq \mathcal{O}(\sqrt{N}). \tag{4}$$

*Proof.* *See Proposition 3 in the Appendix for the full derivation.*

This establishes that rank collapse is intrinsic for Task Arithmetic-based model merging techniques, regardless of the merging coefficient $\alpha$. Consequently, **standard Task Arithmetic methods cannot prevent the marginalization of task-specific information, regardless of the chosen merging coefficient.** In other words, any Task Arithmetic-based method will inevitably emphasize the common components while disregarding the task-specific information as more models are merged. This can also be seen in Fig. 3, as the common singular values scale linearly compared to the tail. For empirical evidence, refer to Section E.4 in the Appendix.

## 4 MODEL MERGING FROM A DECOMPOSED SUBSPACE PERSPECTIVE

Building on our prior analysis of rank collapse as well as previous works linking rank collapse (Dziadzio et al., 2025; Milbich et al., 2020) to reduced generalization, we propose two solutions. Firstly, we introduce *Subspace Boosting* (Sec. 4.1) to directly mitigate rank collapse in task vectors. Furthermore, we leverage HO-GSVD (Sec. 4.2) to create an interpretable framework for model merging and expert selection.

### 4.1 *Subspace Boosting* FOR MODEL MERGING

To directly mitigate the rank collapse issue identified in Sec. 3, we introduce *Subspace Boosting*, a general method that operates on merged task vectors, ensuring compatibility with modern merging techniques that rely on task vectors (Yadav et al., 2023; Ilharco et al., 2023a; Wang et al., 2024b) (see pseudocode provided in Alg. 1).

We posit that preventing rank collapse is essential for effective merging. As more experts are merged, this collapse forces the models to encode an increasing amount of information into progressively constrained task vector subspaces, harming generalization. *Subspace Boosting* effectively recovers the task-specific information lost during merging, which is proven for averaging in Prop. 4, improving the merging process efficacy and final model performance.

*Subspace Boosting* is applied to the merged task vector from any Task Arithmetic-based method such as TA, TIES, or Consensus merging. Each weight matrix in the merged task vector (denoted by "param" in Alg. 1) is independently decomposed through the following steps. (i) Initially, the weight matrix is decomposed via SVD. (ii) Afterwards, the hyperparameter $\beta$ (denoted by "beta" in Alg. 1) is used to determine a cutoff point for the cumulative sum of singular values. All singular values smaller than the one at the cutoff index are "boosted" by clamping them to the cutoff value, visualized in Fig. 1b. (iii) Finally, the new weight matrix is reconstructed using the original singular vectors, but with the new, boosted singular values.

As observed, *Subspace Boosting* requires only one hyperparameter, namely the boosting threshold $\beta$, which determines the cutoff after which the smaller singular values are boosted. Across our experiments, our findings indicate that $\beta$ is highly robust, with minimal tuning required. Also, *Subspace Boosting* provides superior performance at 6x faster wall clock time over state-of-the-art methods (Gargiulo et al., 2025) (results in Supplementary Sec. C).

In addition, Fig. 1a demonstrates that *Subspace Boosting* effectively mitigates the rank collapse issue in task vectors by utilizing the full subspace (of dimensionality 768 for ViT-B/16 models), compared to other existing Task Arithmetic-based methods.

**Algorithm 1:** Pytorch style pseudocode for *Subspace Boosting*

```python
def subspace_boosting(param, beta):
    """
    param: weight matrix
    beta: boosting threshold
    """
    U, S, Vh = svd(param)
    t_sum = S.sum()
    c_sum = torch.cumsum(S, dim=0)
    n_sum = c_sum / t_sum
    k = (n_sum >= beta).nonzero()
    idx = k[0].item()
    S_new = torch.clamp(S, S[idx])
    new_param = U @ diag(S_new) @ Vh

    return new_param
```

### 4.2 BREAKING DOWN TASK VECTORS WITH HIGHER ORDER GENERALIZED SVD

While *Subspace Boosting* successfully recovers the lost information, it operates indiscriminately on all singular values, without identifying ***common*** or ***unique*** subspaces. To solve this, we move from SVD to a framework that allows direct comparison between task vectors.

We therefore leverage **Higher Order Generalized SVD (HO-GSVD)**, a technique that projects multiple matrices into a single, shared subspace. This enables a straightforward comparison of multiple expert weights. As proposed in Ponnapalli et al. (2011); Kempf et al. (2023); Loan (1976); Golub & Van Loan (2013), given a set of $N$ matrices $A_1, \ldots, A_N$, HO-GSVD decomposes each matrix $A_i$ as:

$$A_i = U_i \Sigma_i V^T, \quad i = 1, \ldots, N, \quad (5)$$

resulting in *distinct* $U_i \in \mathbb{R}^{m_i \times n}$, $\Sigma_i \in \mathbb{R}^{n \times n}$ and a *shared* subspace $V \in \mathbb{R}^{n \times n}$, identical for all factorizations. In our case, the matrices $A_i$ correspond to the

**HO-GSVD**

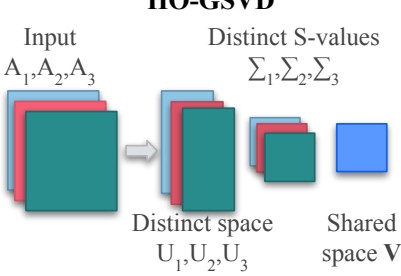

Figure 4: **Higher-Order Generalized SVD (HO-GSVD).** Unlike normal Singular Value Decomposition (SVD) which decomposes matrices into individual $A_i = U_i \Sigma_i V_i$, HO-GSVD allows for decompositions into shared right singular subspaces $V$.

weight matrices originating from the independent task vectors $1 \leq i \leq n$, for a certain layer. By establishing a shared subspace $V$, this decomposition enables the direct comparison of different models and identification of **common** or **unique** subspaces for different tasks. The complete details are provided in Supplementary Sec. G.

**Matrix Decomposition Comparison via Alignment Matrices** To evaluate the significance of dimension $V_k$ (of the shared space) for one matrix relative to another, we take the ratio of their corresponding generalized singular values $\sigma_{i,k}/\sigma_{j,k}$. A ratio close to 1 indicates a **common** subspace, while a ratio deviating highly from 1 suggests that the dimension is more **unique** or important to one of the matrices. This allows us to express each matrix as a sum of common and unique components:

$$A_i = \underbrace{\sum_{k \in \mathcal{I}_{>}1} \sigma_{i,k} u_{i,k} v_k^T}_{\text{common}} + \underbrace{\sum_{k \in \mathcal{I}_1} \sigma_{i,k} u_{i,k} v_k^T}_{\text{unique}}, \tag{6}$$

where $\mathcal{I}_{>}1$ and $\mathcal{I}_1$ denote the common and unique subspaces, respectively.

To quantify the degree of interference between two matrices, we introduce the *Alignment Matrix* $\mathbf{A} \in \mathbb{R}^{N \times N}$. We define matrices as *well-aligned* if they rely on different subspaces (low interference) and *poorly aligned* if they share important subspaces (high interference). The alignment between matrices $A_i$ and $A_j$ is calculated as the average log ratio of their generalized singular values across all $L$ weight matrices from the respective layers:

$$(\mathbf{A})_{ij} = \frac{1}{L} \sum_{l=1}^{L} \left( \frac{1}{M_l} \sum_{p=1}^{M_l} \left| \log \left( \frac{\sigma_{i,p}^{(l)} + \epsilon}{\sigma_{j,p}^{(l)} + \epsilon} \right) \right| \right) \tag{7}$$

where $\sigma_{i,p}^{(l)}$ is the $p$-th generalized singular value of model $i$ for $l$-th weight matrix, and $M_l$ is the number of generalized singular values for that matrix, with $\epsilon = 1e^{-12}$ for stability. Unlike other methods such as direct weight comparisons, this approach offers a novel way to *compare task vectors within a shared, decomposed subspace*. This allows us to optimize subspace overlap to select diverse, low interference experts.

Finally, we also introduce *Higher-Order Subspace Boosting* by replacing SVD with HO-GSVD in *Subspace Boosting*. This is the first interpretable model merging method, achieving strong performance. For further details, please refer to Supplementary Sec. G.

## 5 EXPERIMENTS

**Baselines and Datasets.** As described in Sec. 4.1, we apply *Subspace Boosting* to the task vectors obtained from several state-of-the-art model merging techniques. The foundational method, Task Arithmetic (Ilharco et al., 2023a), defines task vectors and merges them via simple averaging. Building on this, TIES-Merging (Yadav et al., 2023) reduces interference by pruning low-magnitude weights from each vector before performing a sign-aware averaging. Consensus Merging (Wang et al., 2024b) further refines this by only keeping weights that are important for at least two of the tasks being merged. We also evaluate compatibility with *LiNeS* (Wang et al., 2025), an orthogonal post-processing technique that scales updates based on layer depth.

*Datasets.* We follow the previous state-of-the-art methods (Wang et al., 2024b; 2025) and evaluate our model merging techniques on image classification tasks, considering the same grouping of 8, 14, and 20 tasks. For language tasks, we follow Tam et al. (2024a) and evaluate on 8 QA tasks, presented by Zhou et al. (2023) and 7 NLP tasks, provided by Yadav et al. (2023).

**Implementation Details.** Following previous works (Ilharco et al., 2023a; Wang et al., 2024b; 2025; Yadav et al., 2023), we employ the CLIP model (Radford et al., 2021a) with ViT-B/32, ViT-B/16 and ViT-L/14 as vision encoders. We use the pretrained and finetuned checkpoints provided by Wang et al. (2024b) and utilize the official code provided by the original authors for all model merging techniques, including the official hyperparameters. *Subspace Boosting* requires only one hyperparameter $\beta$ that determines the number of dimensions to be boosted. We tune $\beta$ on the validation set by performing a simple search over the set $\{0, 0.01, 0.02\}$. The updated task vector components of the Transformer architecture (Dosovitskiy et al., 2021) are the linear layers and the attention layers. For the language domain, we utilize T5 transformers (Raffel et al., 2020), provided by Tam et al. (2024a) and apply our method to the same layers.

Table 1: ***Subspace Boosting*** **significantly improves model merging efficacy.** Accuracy performance results (in %) for merging vision classification benchmarks with 8, 14 and 20 tasks (Wang et al., 2024b) when *Subspace Boosting* is applied to Task Arithmetic (TA) (Ilharco et al., 2023a;a), TIES-Merging (Yadav et al., 2023), Consensus Merging (Wang et al., 2024b), or LiNeS (Wang et al., 2025). Best performing result in each group is indicated in bold, while the second best is underlined.

| Method | LiNeS | ViT-B/32 | | | ViT-B/16 | | | ViT-L/14 | | |
|---|---|---|---|---|---|---|---|---|---|---|
| | | 8 tasks | 14 tasks | 20 tasks | 8 tasks | 14 tasks | 20 tasks | 8 tasks | 14 tasks | 20 tasks |
| Zero-Shot | – | 48.3 | 57.3 | 56.1 | 55.5 | 61.4 | 59.8 | 64.8 | 68.3 | 65.3 |
| Finetuned | – | 90.5 | 89.5 | 90.4 | 92.6 | 91.6 | 92.3 | 94.0 | 93.3 | 94.0 |
| Task Arithmetic | ✗ | 69.7 | 65.0 | 60.3 | 74.6 | 70.4 | 65.7 | 84.0 | 79.2 | 74.0 |
| | ✓ | 74.2 | 69.1 | 63.4 | 77.6 | 72.7 | 67.7 | 86.5 | 82.2 | 77.1 |
| + *Subspace Boosting* (**ours**) | ✗ | 83.1 | 75.8 | 66.4 | 87.7 | 82.0 | 71.6 | 91.4 | 86.2 | 80.6 |
| | ✓ | **85.6** | **80.8** | **77.2** | **88.8** | **84.7** | **80.0** | **92.6** | **89.3** | 87.2 |
| TIES-Merging | ✗ | 73.6 | 67.6 | 63.1 | 79.1 | 73.0 | 68.1 | 85.6 | 79.3 | 75.6 |
| | ✓ | 77.2 | 72.1 | 67.2 | 79.9 | 75.2 | 71.2 | 88.0 | 82.5 | 79.6 |
| + *Subspace Boosting* (**ours**) | ✗ | 81.8 | 74.4 | 69.8 | 87.0 | 80.5 | 75.9 | 91.1 | 83.6 | 82.0 |
| | ✓ | 83.8 | 79.1 | 75.9 | 87.4 | 83.3 | 79.7 | 91.9 | 86.1 | 85.9 |
| Consensus Merging | ✗ | 74.5 | 70.1 | 65.3 | 78.9 | 73.9 | 70.2 | 85.2 | 81.9 | 78.7 |
| | ✓ | 77.1 | 73.6 | 68.6 | 79.5 | 75.8 | 72.0 | 87.3 | 84.0 | 81.0 |
| + *Subspace Boosting* (**ours**) | ✗ | 82.7 | 77.1 | 73.2 | 87.0 | 81.9 | 77.6 | 91.5 | 86.4 | 84.9 |
| | ✓ | 84.4 | 80.3 | **77.2** | 87.6 | 84.2 | **80.0** | 92.2 | 88.8 | **87.9** |

## 5.1 *Subspace Boosting* ENHANCES PERFORMANCE ACROSS VISION AND LANGUAGE TASKS

Figure 1a illustrates the impact of *Subspace Boosting* on the stable rank scores of task vectors obtained from TA, TIES and Consensus Merging. Notably, *Subspace Boosting* successfully utilizes the available weight space by increasing the matrix rank. This enables the models to better populate the corresponding subspaces and mitigates rank collapse as more and more experts are incorporated. Additional results are provided in Supplementary Sec D.

**Vision Results.** As shown in Table 1, *Subspace Boosting* significantly improves performance of standard merging techniques across 8, 14 and 20 vision tasks. By enabling the methods to leverage a higher effective subspace, our approach yields substantial gains for all methods; for example it boosts the accuracy of simple Task Arithmetic from 65.0% to 75.8% when merging 14 tasks. Notably, this elevates all baselines to comparable performance.

Table 2: **Language Results.** Performance comparison (in terms of accuracy in %) on two language task collections. Our *Subspace Boosting* variant with TA and LiNeS achieves the best performance on both benchmarks.

| Method | 8 Tasks | 7 Tasks |
|---|---|---|
| Zero-shot | 33.1 | 44.9 |
| Fine-tuned | 80.7 | 85.9 |
| Task Arithmetic (TA) | 63.8 | 71.9 |
| TIES Merging | 63.0 | 71.6 |
| Consensus Merging | 68.6 | 73.5 |
| TA + LiNeS | 67.6 | 76.4 |
| Subspace Boosting (**ours**) | **75.3** | **83.0** |

To demonstrate its generality, we show that *Subspace Boosting* also enhances orthogonal techniques such as LiNeS. When merging 14 ViT-B/32 experts, LiNeS alone improves Task Arithmetic's performance from 65.0% to 69.1%, whereas applying *Subspace Boosting* provides a further, substantial improvement to 80.8%, highlighting its complementarity. Overall, the performance gains remain substantial across methods (as illustrated in Fig. 1a), model sizes, and expert model counts.

**Language Results.** We also evaluate *Subspace Boosting* with TA and LiNeS using popular language benchmarks. For 8 QA tasks (Zhou et al., 2023), *Subspace Boosting* shows the same significant improvements as in the vision domain, improving TA by around 12% when applied with LiNeS. For 7 NLP tasks, provided by Yadav et al. (2023), the improvements are of similar magnitude.

## 5.2 ABLATIONS

To better understand the behavior of *Subspace Boosting*, we conduct ablation studies reporting the results in Tab. 3. The reported results use 8 ViT-B/16 models merged with TA (Ilharco et al., 2023a).

Table 3: *Subspace Boosting* **Ablations.** We analyze the impact of the boosting threshold $\beta$, the layer type, and layer-specific $\beta$. The results are reported for 8 ViT-B/16 models merged via TA (Ilharco et al., 2023a). (a) *Subspace Boosting* is highly robust to the choice of $\beta$, (b) performs best when applied to all layers, and (c) requires no tuning when modifying both weight types.

(a) **Boosting threshold**. *Subspace Boosting* demonstrates robustness to variations of the $\beta$ value.

| $\beta$ | Accuracy (%) |
|---|---|
| 0.00 | 87.7 |
| 0.01 | 87.4 |
| 0.02 | 87.2 |

(b) **Layers**. We observe that both the fully connected (FC) and the attention (Attn) layers contribute to the performance gain.

| Layer | Accuracy (%) |
|---|---|
| FC | 86.5 |
| Attn | 83.9 |
| Both | 87.7 |

(c) **Boosting threshold across layers**. In this setting, the attention (Attn) and fully connected (FC) share the same optimal $\beta$.

| FC | Attn | Accuracy (%) |
|---|---|---|
| 0.00 | 0.00 | 87.7 |
| 0.00 | 0.01 | 87.6 |
| 0.01 | 0.01 | 87.4 |
| 0.02 | 0.01 | 87.4 |

Table 4: **Comparison to State-of-the-Art.** Our best-performing *Subspace Boosting* variant with LiNeS achieves results comparable to other state-of-the-art methods, but at a fraction of the computational overhead and complexity.

| Method | ViT-B/32 | | | ViT-B/16 | | | ViT-L/14 | | |
|---|---|---|---|---|---|---|---|---|---|
| | 8 tasks | 14 tasks | 20 tasks | 8 tasks | 14 tasks | 20 tasks | 8 tasks | 14 tasks | 20 tasks |
| Finetuned | 90.5 | 89.5 | 90.4 | 92.6 | 91.6 | 92.3 | 94.0 | 93.3 | 94.0 |
| TSV-M[†] (Gargiulo et al., 2025) | 83.8 | 79.5 | 76.7 | 87.2 | 83.7 | 80.3 | 91.2 | 88.3 | 87.3 |
| Iso-C[†] (Marczak et al., 2025) | 83.8 | 79.1 | 74.7 | 88.6 | 83.3 | 79.0 | 92.4 | 88.2 | 87.0 |
| Iso-CTS[†] (Marczak et al., 2025) | 83.8 | 80.2 | 77.3 | **89.1** | 85.0 | **81.6** | **93.0** | **89.6** | **89.3** |
| TA + *Subspace Boosting* (**ours**) | **85.6** | **81.7** | **77.6** | 88.9 | **85.1** | 81.1 | 92.7 | 89.4 | 88.0 |

Our analysis investigates three key aspects. Firstly, we investigate the robustness to the boosting parameter $\beta$. As shown in Table 3a, our method is robust to variations of $\beta$, since performance only fluctuates between 87.2% and 87.7% when $\beta \in [0.00, 0.01, 0.02]$. Secondly, we examine the contributions of different layer types, seen in Table 3b. We observe that both the attention and fully connected (FC) layers contribute to improving performance, with the best accuracy of 87.7% achieved when applying *Subspace Boosting* to the full model. Finally, we analyze whether attention layers require a different value for $\beta$ than FC layers. As reported in Table 3c, for *Subspace Boosting*, the attention and FC layers share the same optimal $\beta$ value. This reveals that *Subspace Boosting* can be applied across both weight types without additional tuning.

**State-of-the-Art Comparison.** We compare our method against recent state-of-the-art techniques (Gargiulo et al., 2025; Marczak et al., 2025) in Table 4. We equivalently tune the merging coefficient $\alpha$ over 30 interpolation points. The results demonstrate that simple TA enhanced with LiNeS and *Subspace Boosting* surpasses both TSV-M and Iso-C and matches or surpasses Iso-CTS. This is particularly evident for ViT-B/32 models, outperforming Iso-CTS by almost 2% for 8 models. However, unlike the other methods, *Subspace Boosting* is both compatible with all other TA based methods as well as over 6x more computationally efficient (details in Supplementary Sec. C).

## 5.3 INTERPRETABLE *Subspace Boosting* VIA HO-GSVD

**Comparison of Generalized Singular Values.** A key limitation of modern model merging is its lack of interpretability. To address this, we leverage HO-GSVD to enable a transparent investigation of the merging process and to compare contributions of individual experts. Fig. 5a illustrates this by visualizing the distribution of generalized singular values for an exemplary weight matrix of a ViT-B/16 attention block projection layer. We compare the generalized singular value distribution of 4 task vectors (Cars, DTD, Eurosat, GTSRB), against the averaged task vector over 8 tasks.

Since HO-GSVD operates on a shared decomposition space, we can directly compare generalized singular values across tasks. The plot reveals a large proportion of shared generalized singular vectors among the tasks, implying that the top dimensions experience high interference. Furthermore, this visualization clearly illustrates rank collapse without needing additional metrics. For the merged model, a significant number of singular values (in the 0-200 index range) are near zero. In contrast, the generalized singular values for the independent task vectors remain well above this floor (Thresh-

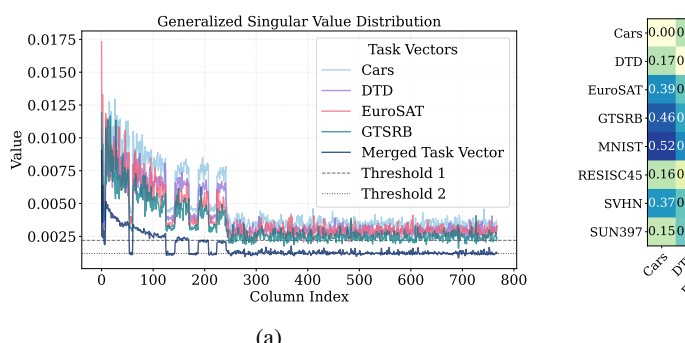
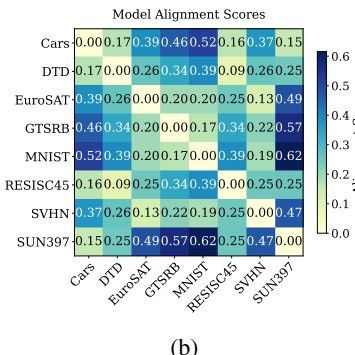

(a)            (b)

Figure 5: **(a) Distribution of Generalized Singular Values** across different task vectors and a merged reference of eight task vectors. **(b) Model Alignment via HO-GSVD**, showcasing how HO-GSVD can be used to contrast expert alignments across a shared decomposition space.

old 1), indicating that the merged task vector occupies a lower-rank subspace. Finally, we observe that the merged task vector is significantly below the others, implying that the used merging coefficient $1/N$ for the average is suboptimal, pointing to a new promising direction for future research: using HO-GSVD to *automatically* choose the optimal merging coefficient.

**Selecting Optimal Experts For Better Performance.** Another benefit of HO-GSVD is comparing the relative significance of a certain subspace for matrix $A_i$ compared to matrix $A_j$, as described in Sec. 4.2. This can be used to select the optimal set of models for merging, which is a combinatorial complex problem (e.g. selecting 6 out of 20 leads to nearly 40,000 possible permutations).

HO-GSVD enables us to utilize the previously defined *Alignment Matrix*, where the higher the value for a pair of models $(A_i, A_j)$, the easier it is to merge both models with reduced interference. An example of an *Alignment matrix* is presented in Fig. 5b. Higher scores correspond to models that are more well aligned (easier to merge).

To demonstrate the effectivenss of the *Alignment Matrix* for 20 ViT models (more results in Supplementary Sec. H) for expert selection, we design an experiment to construct a merged model of 8 experts. Three of these experts are pre-selected to cover in-distribution (Pool) tasks. The remaining five are chosen from 14 candidates either via a random baseline (averaged over 10 draws) or our proposed method, which selects models that minimize interference with the pre-selected Pool tasks.

Table 5: **HO-GSVD facilitates expert selection.** Choosing experts from a larger pool via HO-GSVD improves/-maintains transfer to candidate tasks (*Pool*) *while* maintain transfer to external (*Ext*).

| Method | Accuracy | |
|---|---|---|
| | **Pool** | **Ext** |
| Random | 72.9 | 47.6 |
| HO-GSVD (ours) | 75.7 | 47.6 |

The final merged models are evaluated on the Pool tasks as well as a separate set of 3 out-of-distribution (OOD) tasks. As shown in Table 5, our proposed method improves in-distribution performance, boosting accuracy from 72.9% to 75.7% while maintaining performance on OOD tasks. This result demonstrates the efficacy of applying HO-GSVD for model selection.

## 6 CONCLUSION

In this work, we discovered and proved *rank collapse* as a fundamental limitation of Task Arithmetic model merging methods (Ilharco et al., 2023b; Yadav et al., 2023; Wang et al., 2024c). Consequently, we proposed *Subspace Boosting* to mitigate this limitation, thereby achieving significant performance improvements that exceed 10% in a wide range of settings across vision and language domains. Additionally, we provide a novel framework using HO-GSVD to address the black-box nature of modern model merging techniques. Moreover, we demonstrated that by comparing shared task subspaces via the *Alignment Matrix*, we can effectively evaluate the behavior of model merging and offer a strong approach to select a subset of models that achieve higher performance when merged.

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

## Appendix Table of Contents

## A  LIMITATIONS

Current Task Arithmetic-based methods, including *Subspace Boosting* and *Higher-Order Subspace Boosting*, require tuning the best merging coefficient. This requires access to a reasonable validation dataset as well as compute resources in order to tune this hyperparameter. In future work, we consider automating this and removing the necessity for tuning. Another limitation is that as the number of merged models grows, the performance continues to decrease. Therefore, finding optimal methods to prevent this degradation could be another interesting direction for future work.

## B  ADDITIONAL IMPLEMENTATION DETAILS

**Hyperparameters**. The results presented with *Subspace Boosting* are obtained with $\beta$ optimized over the small set $\{0.00, 0.01, 0.02\}$. Given the robustness, for *Higher-Order Subspace Boosting*, we simply kept $\beta = 0.0$. In regards to merging coefficients, we utilized the same range as provided by (Wang et al., 2024c) for all our results $\alpha \in \{0.1, 0.2, ..., 1.0\}$. For additional baseline methods, such as TSV-M (Gargiulo et al., 2025) and Iso-C, Iso-CTS (Marczak et al., 2025), we extend the range to $\alpha \in \{0.1, 0.2, ..., 3.0\}$, as utilized by the original authors, and a range of 30 of $\alpha \in \{0.1, ..., 0.5\}$ for *Subspace Boosting*. We benchmark TSV-M, Iso-C, and Iso-CTS using the previously provided checkpoints, which are slightly inferior to the checkpoints used by (Gargiulo et al., 2025) and Iso-CTS (Marczak et al., 2025). For Consensus Merging (Wang et al., 2024b), all results are evaluated using Task Arithmetic as the baseline method.

**Models and Datasets.** To ensure direct replicability and comparability, we extend the repository provided by Wang et al. (2024c), and use the same baseline checkpoints, finetuned checkpoints and datasets as the authors. For the 8-, 14- and 20-task benchmarks, we use the same datasets as the

original authors. For the language domain, we use the repository provided by Tam et al. (2024a) and the respective models and datasets.

## C COMPUTATIONAL RESOURCES

For all of our experiments, we leverage a compute cluster equipped with 2 NVIDIA GeForce RTX 2080 Ti with 12 GB VRAM and 6 NVIDIA Quadro RTX 6000 GPUs with 24 GB VRAM; alongside 2 Intel Xeon Silver 416 CPU @ 2.10 Ghz CPUs and 256 GB of RAM.

The running time for our *Subspace Boosting* compared to the state-of-the-art results and the task arithmetic baselines is reported in Tab. 6. *Subspace Boosting* introduces additional computational overhead, however, we notice that the computational overhead for SVD and boosting is very negligible and in line with other, simpler methods. We notice that the computational overhead of our method is in line with the other, simpler methods such as TIES and over 6x more efficient in terms of clock-time than TSV-M and Iso-CTS (which extends TSV-M).

Table 6: Comparison of execution time (in seconds) of *Subspace Boosting* against other methods for 20 tasks. *Subspace Boosting* is 6× faster than TSV-M.

| Method | ViT-B/32 | ViT-B/16 | ViT-L/14 |
|---|---|---|---|
| Task Arithmetic | 0.2s | 0.28s | 1s |
| Consensus Merging | 1.8s | 2.4s | 20.2s |
| TIES | 6.3s | 4.3s | 25.6s |
| *Subspace Boosting* (ours) | 9.7s | 10.5s | 40.1s |
| TSV-M | 62s | 63s | 210s |

## D ADDITIONAL RANK COLLAPSE RESULTS

We also employ the *cumulative energy rank* to measure the rank collapse. The *cumulative energy rank* $\mathcal{M}_{\text{cer}}$ measures how much of the matrix information is captured by the top singular values, or more precisely how many singular values are required to cover $k\%$ of the energy $\mathcal{E} := \sum_i^n \sigma_i^2$.

We visualize rank collapse using both metrics, namely the stable rank and the cumulative energy rank. In Figure 6, the stable rank is plotted across different layers and components. Rank collapse is evident across the majority of the layers and components. For the projection layer of the MLP component, rank collapse is especially evident across all visualized layers.

Figure 7 shows the stable rank and the normalized value over an increasing number of merged models (4, 8, 14, 20) via Task Arithmetic (Ilharco et al., 2023a). To calculate the normalized values, each component's values are divided by the largest value for the respective component. This keeps the range (0-1) identical across all layers for easier comparison. As more models are merged, the stable rank clearly decreases in all layers and for almost all components. Although it would be expected for a merged model to maintain maximal rank as it must account for multiple tasks simultaneously, due to the rank collapse, naively merged models often rely on increasingly smaller subspaces for the classification of increasing number of tasks. This hinders the model's generalization capability as the task becomes more complex and higher in number.

We also employ the cumulative energy as a metric for the intrinsic dimension of the model across layers to provide another perspective on rank collapse. In Figure 8, the number of components necessary to sum up to 50% of the energy (cumulative energy rank) is visualized. We observe a similar trend to the previous results, and observe that for a vast majority of layers and components the cumulative energy rank decreases as we merge more models.

Moreover, we observe a direct correlation between the performance of various model merging methods and their respective cumulative energy rank (see Figure 9 for 14 merged models). For detailed performance reports, please refer to Table 1. It is noticeable that Task Arithmetic merging ranks are often small or negligible, and while TIES does maintain a higher cumulative energy rank, Consensus TA merging consistently achieves a higher rank - which is tied with increasing overall merging performance.

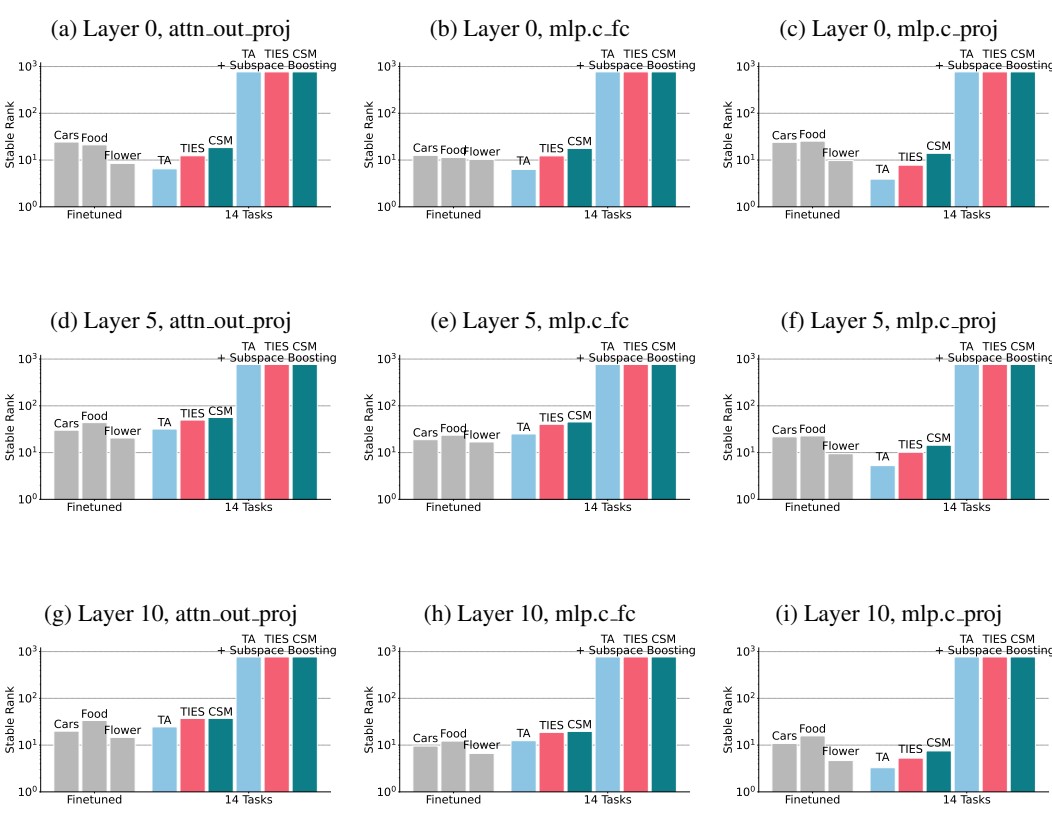

Figure 6: Stable rank visualized across multiple layers and different components. It is noticeable that the baseline methods TA, TIES and CSM exhibit small stable rank values, however by applying *Subspace Boosting* the stable rank score increases considerably. We report the layer and the component on top of each subplot.

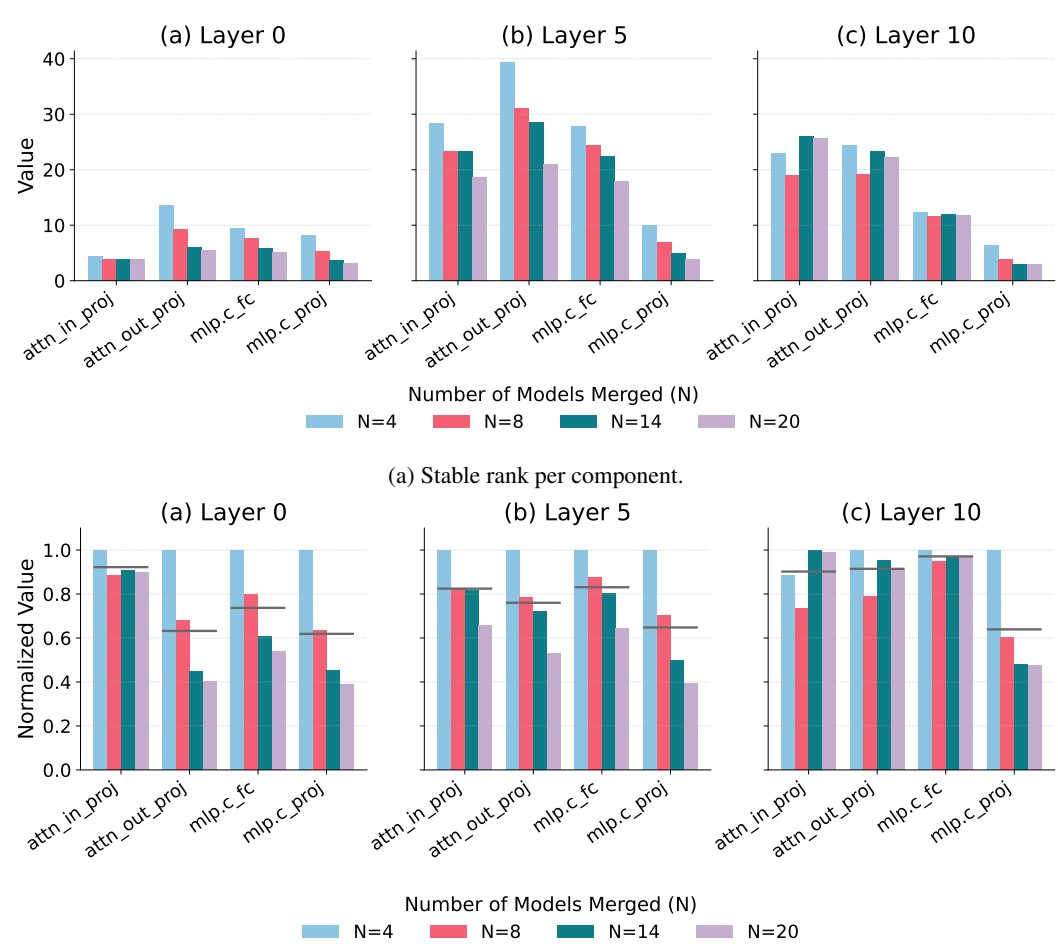

(a) Stable rank per component.

(b) Normalized stable rank per component. The mean is plotted in gray.

Figure 7: Absolute and normalized stable rank across a different number of merged models via Task Arithmetic. It is noticeable that merging more models decreases the rank of task vectors, limiting the expression space of the model.

# E    PROOFS FOR RANK COLLAPSE AND SUBSPACE BOOSTING

In this section, we provide the derivations for the previously stated propositions. We first prove that *rank collapse* is inevitable for Task Arithmetic-based merging methods under the mild assumption of orthogonality between task-specific information. Subsequently, we show that Subspace Boosting boosts essential task-specific information that disappears as more models are merged.

For clarity and rigor, we first provide the formal restatement of each introduced proposition before presenting the corresponding proof.

## E.1    RANK COLLAPSE AND SPECTRAL DECAY OF SINGULAR VALUES CORRESPONDING TO TASK-SPECIFIC INFORMATION UNDER AVERAGING

To theoretically analyze the phenomenon of rank collapse and spectral decay (the rate at which the singular values decay), we model the averaged task vector $\Delta_m$ obtained from merging $N$ models via simple Task Arithmetic with the merging coefficient $\alpha = 1/N$. For simplicity, we focus on only one layer and the corresponding weight matrices. We decompose the averaged matrix into a *common* component, representing shared information across the individual matrices, and a *unique*

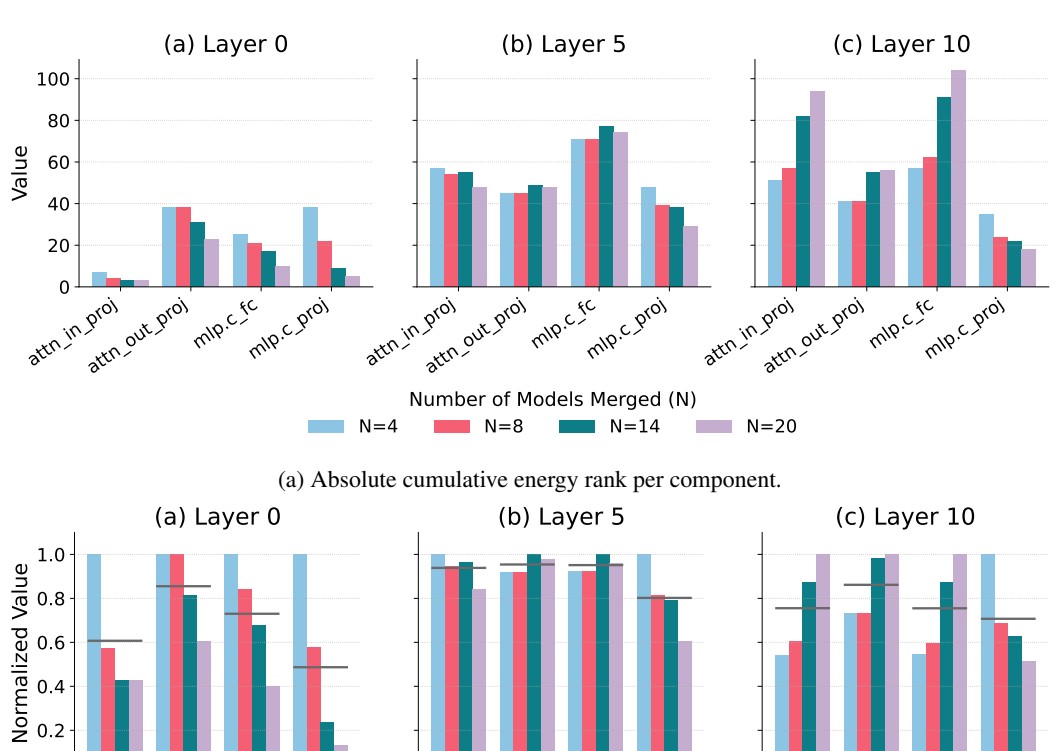

(a) Absolute cumulative energy rank per component.

(b) Normalized cumulative energy rank per component. The mean is plotted in gray.

Figure 8: Absolute and normalized cumulative energy rank for 50% of the energy when Task Arithmetic is employed. It is noticeable that the cumulative energy rank decreases with the increase of the number of merged tasks in a majority layers, especially earlier ones, leading to rank collapse.

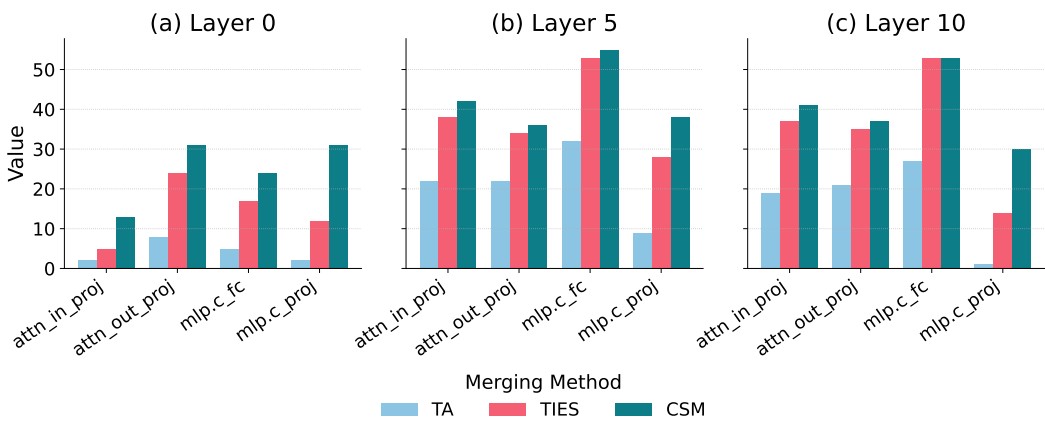

Figure 9: Cumulative energy rank for 30% of the energy for different merging methods when 14 tasks are merged.

component, representing either task-specific variations or noise:

$$\Delta_m = \Delta_{\text{common}} + \Delta_{\text{unique}}. \tag{8}$$

**Task-Specific Information:** Let $\Delta_{\text{unique}} = \frac{1}{N} \sum_{i=1}^{N} (U_i + \mathcal{E}_i)$. The matrices $U_i$ represent task-specific components. Crucially, we assume that $U_i$ are *pairwise orthogonal* in the parameter space. That is, if we view the matrices as flattened vectors, they are orthogonal. Formally, this corresponds to orthogonality with respect to the Frobenius inner product: $\langle U_i, U_j \rangle_F = 0$ for $i \neq j$. Subsequently, we model random noise as $\mathcal{E}_i \sim \mathcal{N}(0, \sigma^2 I)$, sampled from an isotropic Gaussian distribution with independent and identically distributed entries. We treat the corresponding matrices $\mathcal{E}_i$ also as flattened vectors. Due to its random nature, this noise is also orthogonal.

**The Common Information:** Let $\Delta_{\text{common}} = \frac{1}{N} \sum_{i=1}^{N} C_i$. This term captures the shared and common components across all matrices. We analyze the total energy (squared Frobenius norm) of this average to determine the behavior of the top $k$ singular values ($\sigma_1 \ldots \sigma_k$), which form the spectral *spike* seen in Figure 3.

**Proposition 1** (Formal Statement: Singular Value Decay of Averaged Task-Specific Information)**.** *Let $\{\Delta_i\}_{i=1}^{N}$ be a set of $N$ task vectors, decomposed into common and unique components. Assume the task-specific components $E_i = U_i + \mathcal{E}_i$, where $B$ can be set to the maximum Frobenius norm across the set of task-specific components $E_i$, such that the task-specific components are bounded in energy as $\|E_i\|_F \leq B$, and are pairwise orthogonal in the parameter space, satisfying:*

$$\langle E_i, E_j \rangle_F = 0 \quad \text{for } i \neq j. \tag{9}$$

*Let $\Delta_{unique}$ be the averaged task-specific matrix:*

$$\Delta_{unique} = \frac{1}{N} \sum_{i=1}^{N} E_i = \frac{1}{N} \sum_{i=1}^{N} (U_i + \mathcal{E}_i). \tag{10}$$

*Then, as $N \to \infty$, the spectral norm (largest singular value) of the averaged task-specific component decays to zero at a rate of $\mathcal{O}\left(\frac{1}{\sqrt{N}}\right)$:*

$$\sigma_1(\Delta_{unique}) = \|\Delta_{unique}\|_2 \leq \mathcal{O}\left(\frac{1}{\sqrt{N}}\right). \tag{11}$$

*Proof.* We analyze the squared Frobenius norm (total energy) of the averaged task-specific matrix. Using the linearity of the sum and the inner product definition of the squared norm:

$$\|\Delta_{\text{unique}}\|_F^2 = \left\| \frac{1}{N} \sum_{i=1}^{N} E_i \right\|_F^2 = \frac{1}{N^2} \sum_{i=1}^{N} \sum_{j=1}^{N} \langle E_i, E_j \rangle_F. \tag{12}$$

By the assumption of pairwise orthogonality, the cross-terms $\langle E_i, E_j \rangle_F$ vanish for all $i \neq j$. The summation therefore simplifies to only the diagonal terms ($i = j$):

$$\|\Delta_{\text{unique}}\|_F^2 = \frac{1}{N^2} \sum_{i=1}^{N} \|E_i\|_F^2. \tag{13}$$

Using the bound $\|E_i\|_F \leq B$, we obtain:

$$\|\Delta_{\text{unique}}\|_F^2 \leq \frac{1}{N^2} (N \cdot B^2) = \frac{B^2}{N}. \tag{14}$$

Taking the squared root yields the decay rate for the Frobenius norm: $\|\Delta_{\text{unique}}\|_F \leq \frac{B}{\sqrt{N}}$. Finally, utilizing the norm inequality $\|A\|_2 \leq \|A\|_F$, it follows that:

$$\sigma_1(\Delta_{\text{unique}}) = \|\Delta_{\text{unique}}\|_2 \leq \|\Delta_{\text{unique}}\|_F \leq \frac{B}{\sqrt{N}}. \tag{15}$$

Thus, the singular values of the task-specific components decay to zero as $N$ increases, while the common components (which do not satisfy the orthogonality condition) sum constructively and persist. $\qquad \square$

> 🔖 This result formally explains the loss of task-specific information during model averaging. The task-specific information $U_i$, which is orthogonal across diverse tasks, is treated mathematically equivalently to noise $\mathcal{E}_i$. The averaging operator suppresses both at the same rate. Consequently, as $N$ increases, the contribution of unique expert knowledge vanishes relative to the common information, leading to rank collapse.

**Proposition 2** (Formal Statement: Asymptotic Stable Rank Collapse). *Let $\mathcal{M}_{stable}(A) = \|A\|_F^2 / \|A\|_2^2$ denote the stable rank. Given the decay of the task-specific (unique) components established in Proposition 1, the stable rank of the merged task vector $\Delta_m$ converges asymptotically to the stable rank of the common subspace:*

$$\lim_{N \to \infty} \mathcal{M}_{stable}(\Delta_m) = \mathcal{M}_{stable}(\Delta_{common}). \tag{16}$$

*Proof.* Recall the decomposition $\Delta_m = \Delta_{\text{common}} + \Delta_{\text{unique}}$. From Proposition 1, we established that $\|\Delta_{\text{unique}}\|_F \leq \frac{B}{\sqrt{N}}$. Using the norm inequality $\|A\|_2 \leq \|A\|_F$, we have:

$$\|\Delta_{\text{unique}}\|_2 \leq \|\Delta_{\text{unique}}\|_F \leq \frac{B}{\sqrt{N}}. \tag{17}$$

Thus, both the spectral norm $\|\Delta_{\text{unique}}\|_2$ and the Frobenius energy $\|\Delta_{\text{unique}}\|_F^2$ vanish as $N \to \infty$.

We now prove that the norms of the merged task vector converge to the norms of the common component. Applying the reverse triangle inequality $|\|A\| - \|B\|| \leq \|A - B\|$, we obtain:

$$\lim_{N \to \infty} |\|\Delta_m\|_F - \|\Delta_{\text{common}}\|_F| \leq \lim_{N \to \infty} \|\Delta_{\text{unique}}\|_F = 0, \tag{18}$$

$$\lim_{N \to \infty} |\|\Delta_m\|_2 - \|\Delta_{\text{common}}\|_2| \leq \lim_{N \to \infty} \|\Delta_{\text{unique}}\|_2 = 0. \tag{19}$$

This implies that convergence of both norms of the merged task vector:

$$\lim_{N \to \infty} \|\Delta_m\|_F = \|\Delta_{\text{common}}\|_F \quad \text{and} \quad \lim_{N \to \infty} \|\Delta_m\|_2 = \|\Delta_{\text{common}}\|_2. \tag{20}$$

Substituting these limits into the stable rank definition yields:

$$\lim_{N \to \infty} \mathcal{M}_{\text{stable}}(\Delta_m) = \frac{\|\Delta_{\text{common}}\|_F^2}{\|\Delta_{\text{common}}\|_2^2} = \mathcal{M}_{\text{stable}}(\Delta_{\text{common}}). \tag{21}$$

Thus, the effective rank of the model collapses to rely solely on the shared subspace, ignoring task-specific information. □

### E.2 Growth Rate of the Common vs Unique Subspaces

In the previous subsection E.1, we prove that the task-specific information decays under averaging. However, this holds when the merging coefficient $\alpha = 1/N$ is dependent on the number of models merged ($N$). In this subsection, ***we prove that the common subspace dominates the task-specific subspace, regardless of the merging coefficient***.

**Proposition 3** (Formal Statement: Inherent Limitation of Task Arithmetic-Based Merging). *Let the merged task vector be defined as $\Delta_m = \alpha \sum_{i=1}^{N} \Delta_i$, where $\alpha > 0$ is the merging coefficient. Let us decompose each task vector into a common component $C_i$ and a unique component $U_i$, such that $\Delta_i = C_i + U_i$.*

*Let us we view the common components $C_i$ as flattened vectors that have positive mean pairwise cosine similarity $\rho > 0$ and bounded energy $\mu_{\min} \leq \|C_i\|_F \leq \mu_{\max}$. Assume the unique components $U_i$ are orthogonal with bounded energy $\|U_i\|_F \leq B$ and $\langle U_i, U_j \rangle = 0$ for $i \neq j$.*

*As $N \to \infty$, the ratio of the spectral magnitude of the common subspace to the unique subspace diverges at a rate of $\mathcal{O}(\sqrt{N})$:*

$$\frac{\|\Delta_{common}\|_2}{\|\Delta_{unique}\|_2} \geq \mathcal{O}(\sqrt{N}). \tag{22}$$

*This implies that the common components asymptotically dominate the unique components, regardless of the merging coefficient $\alpha$.*

*Proof.* We analyze the growth rates of the squared Frobenius norms (energy) for the common and unique parts separately.

**1. Growth of the Unique (Task-Specific) Component:** Let $\Delta_{\text{unique}} = \alpha \sum_{i=1}^{N} U_i$. Since the components $U_i$ are pairwise orthogonal, their energies sum additively:

$$\|\Delta_{\text{unique}}\|_F^2 = \alpha^2 \sum_{i=1}^{N} \|U_i\|_F^2 \leq \alpha^2 N B^2. \tag{23}$$

Taking the square root, we get $\alpha B \sqrt{N}$, whose magnitude grows as $\mathcal{O}(\sqrt{N})$.

**2. Growth of the Common Component:** Let $\Delta_{\text{common}} = \alpha \sum_{i=1}^{N} C_i$. We expand the squared norm and substitute the definition of the inner product using cosine similarity $\text{sim}(C_i, C_j)$:

$$\|\Delta_{\text{common}}\|_F^2 = \alpha^2 \sum_{i=1}^{N} \sum_{j=1}^{N} \langle C_i, C_j \rangle = \alpha^2 \sum_{i=1}^{N} \sum_{j=1}^{N} \|C_i\|_F \|C_j\|_F \cdot \text{sim}(C_i, C_j). \tag{24}$$

Let $\bar{\rho}$ denote the mean pairwise cosine similarity across all $N^2$ pairs. Given the assumption of positive cosine similarity ($\bar{\rho} > 0$) and bounded energy ($\|C_i\|_F \geq \mu_{\min}$), the summation is bounded below by:

$$\|\Delta_{\text{common}}\|_F^2 \geq \alpha^2 N^2 \mu_{\min}^2 \bar{\rho}. \tag{25}$$

As $N \to \infty$, the quadratic term $N^2$ dominates. Taking the square root, we get $\alpha N \mu_{\min} \sqrt{\bar{\rho}}$, which leads to the magnitude growing as $\mathcal{O}(N)$.

**3. The Common Subspaces Dominate the Unique (Task-Specific) Subspaces:** Comparing the growth rates of the Frobenius norms directly yields:

$$\frac{\|\Delta_{\text{common}}\|_F}{\|\Delta_{\text{unique}}\|_F} \geq \frac{\alpha N \mu_{\min} \sqrt{\bar{\rho}}}{\alpha B \sqrt{N}} = \frac{\mu_{\min} \sqrt{\bar{\rho}}}{B} \sqrt{N}. \tag{26}$$

This yields a magnitude growth of $\mathcal{O}(\sqrt{N})$. To extend this to the spectral norm, we recall that:

$$\|\Delta_{\text{common}}\|_F^2 = \sum_{i=1}^{r} \sigma_i^2 \leq r \sigma_1^2 = r \|\Delta_{\text{common}}\|_2^2 \tag{27}$$

where $r$ denotes the rank of the common subspace, which implies $\|\Delta_{\text{common}}\|_2 \geq \frac{1}{\sqrt{r}} \|\Delta_{\text{common}}\|_F$. Since $\|\Delta_{\text{common}}\|_F$ grows as $\mathcal{O}(N)$ and the rank $r$ is bounded, the spectral norm $\|\Delta_{\text{common}}\|_2$ must also grow as $\mathcal{O}(N)$.

In contrast, the unique component's spectral norm is upper-bounded by its Frobenius norm: $\|\Delta_{\text{unique}}\|_2 \leq \|\Delta_{\text{unique}}\|_F = \mathcal{O}(\sqrt{N})$. Comparing these spectral growth rates:

$$\frac{\|\Delta_{\text{common}}\|_2}{\|\Delta_{\text{unique}}\|_2} \geq \frac{\mathcal{O}(N)}{\mathcal{O}(\sqrt{N})} = \mathcal{O}(\sqrt{N}). \tag{28}$$

Thus, the common components spectrally dominate the unique components by a factor of $\mathcal{O}(\sqrt{N})$.
$\square$

> 🔖 Thus, this formally shows that regardless of the merging coefficient, under Task Arithmetic-based model merging, the common subspaces will dominate the task-specific subspaces as more models are merged. Hence, the task-specific information will be marginalized compared to the common information.

### E.3 SIGNAL-TO-RATIO OF SUBSPACE BOOSTING

In this subsection, we formally prove under which conditions Subspace Boosting improves model averaging. since Subspace Boosting boosts all singular values after the threshold, this also amplifies potential noise that is located in spectral tail. We therefore theoretically justify the effectiveness of Subspace Boosting.

**Proposition 4** (Formal Statement: Signal-to-Noise Trade-off in Subspace Boosting). *Let the spectral tail ($\sigma_i < \sigma_\beta$) of the averaged task vector be composed of useful signal $\Delta_{signal}$ and stochastic noise $\Delta_{noise}$. Recall that Subspace Boosting increases these neglected singular values to be $\sigma_\beta$. Subspace Boosting reduces the reconstruction error ($\mathcal{L}_{boost} < \mathcal{L}_{TA}$) in Frobenius norm provided that the energy of the signal suppressed by averaging exceeds the energy of the noise amplified by boosting:*

$$\|\Delta_{signal}\|_F^2 > d_{noise} \cdot \sigma_\beta^2, \tag{29}$$

*where $d_{noise}$ is the dimensionality of the noise subspace. This condition is satisfied in deep learning regimes where the task-specific signal (accumulated during training) dominates the stochastic gradient noise.*

*Proof.* We define the reconstruction error $\mathcal{L}$ as the squared Frobenius distance between the averaged task-specific signal $\hat{\Delta}$ and the ideal task vector signal $\Delta_{signal}$ within the spectral tail:

$$\mathcal{L}(\hat{\Delta}) = \|\hat{\Delta} - \Delta_{signal}\|_F^2.$$

We analyze this error for two merging methods: Task Arithmetic ($\hat{\Delta} = \Delta_{TA}$) and Subspace Boosting ($\hat{\Delta} = \Delta_{boost}$).

**Assumption 1: Preservation of Direction via Linear Mode Connectivity.** We rely on the Linear Mode Connectivity (LMC) hypothesis (Frankle et al., 2020; Neyshabur et al., 2020), which posits that fine-tuned models reside within a shared, linearly connected low-loss basin. This geometric structure implies that the merged task vector lies within the valid solution manifold spanned by the experts. Consequently, we assume that the averaging process successfully identifies the optimal *orientation* (the center of the manifold) but fails to estimate the optimal *magnitude*. Thus, while the magnitude is suppressed, the direction remains preserved.

**1. Merging via Task Arithmetic.** Let $\Delta_{TA}$ denote the task vector obtained via standard Task Arithmetic. As established in Proposition 1, averaging suppresses task-specific or noisy components at a rate of $\mathcal{O}(1/\sqrt{N})$. As $N \to \infty$, the magnitude of the Task Arithmetic task vector effectively vanishes in the spectral tail: $\Delta_{TA} \to 0$. Consequently, the reconstruction error $\mathcal{L}_{TA}$ is dominated entirely by the energy of the missing signal:

$$\mathcal{L}_{TA} = \lim_{N \to \infty} \|\Delta_{TA} - \Delta_{signal}\|_F^2 = \|0 - \Delta_{signal}\|_F^2 = \|\Delta_{signal}\|_F^2.$$

**2. Merging via Subspace Boosting.** Let us recall that Subspace Boosting sets singular values in the tail that are less than $\sigma_\beta$ to $\sigma_\beta$. We assume $\sigma_\beta$ is chosen to match the signal magnitude, such that the signal component is restored:

$$\|\Delta_{boost} - \Delta_{signal}\|_F^2 \approx 0.$$

However, this operation indiscriminately boosts the $d_{noise}$ dimensions corresponding to noise. To quantify the error, we utilize the orthogonal projection operator $\mathcal{P}_{noise}$, which projects vectors onto the $d_{noise}$ dimensions that are orthogonal to the true task signal. Since the ideal signal $\Delta_{signal}$ is zero in these dimensions ($\mathcal{P}_{noise}(\Delta_{signal}) = 0$), any energy introduced here is pure error. Therefore, the reconstruction loss is the energy of this boosted noise:

$$\mathcal{L}_{boost} = \|\mathcal{P}_{noise}(\Delta_{boost}) - \mathcal{P}_{noise}(\Delta_{signal})\|_F^2$$
$$= \|\mathcal{P}_{noise}(\Delta_{boost}) - 0\|_F^2 = \sum_{j=1}^{d_{noise}} \sigma_\beta^2 = d_{noise} \cdot \sigma_\beta^2. \tag{30}$$

**3. Comparison.** Boosting yields a strictly lower reconstruction error when $\mathcal{L}_{boost} < \mathcal{L}_{TA}$. Substituting the derived terms yields the condition:

$$d_{noise} \cdot \sigma_\beta^2 < \|\Delta_{signal}\|_F^2.$$

This inequality is supported by the Coherent Gradients hypothesis (Chatterjee, 2020), which demonstrates that the energy of coherent task updates scales quadratically with the number of samples

($\mathcal{O}(n^2)$), whereas orthogonal noise scales only linearly $\mathcal{O}(n)$. Therefore, for any sufficiently trained model, the structural energy of the signal dominates the noise, satisfying the condition. Consequently, under these conditions, boosting the tail recovers the significant energy of $\Delta_{\text{signal}}$ while introducing comparatively negligible variance from the noise dimensions. □

> 🔖 Since the task-specific signal decays to 0 as $N \to \infty$, we must amplify the corresponding singular values to minimize the reconstruction error. Assuming the Coherent Gradients hypothesis holds, the signal scales quadratically while the noise scales linearly. Thus, under these conditions, the benefit of recovering the lost signal significantly exceeds the error introduced by boosting the underlying noise.

### E.4 EMPIRICAL VALIDATION OF PROPOSITION 3

We provide empirical evidence supporting the spectral scaling insights for the common ($\Delta_{\text{common}}$) vs. unique ($\Delta_{\text{unique}}$) components, derived in Proposition 3. Theoretically, the Proposition posits that as the number of merged models $N$ increases, the common subspace (shared information) dominates the unique task-specific subspaces. Specifically, it predicts that common components scale linearly with the number of models ($\mathcal{O}(N)$), whereas unique components scale at a rate of $\mathcal{O}(\sqrt{N})$.

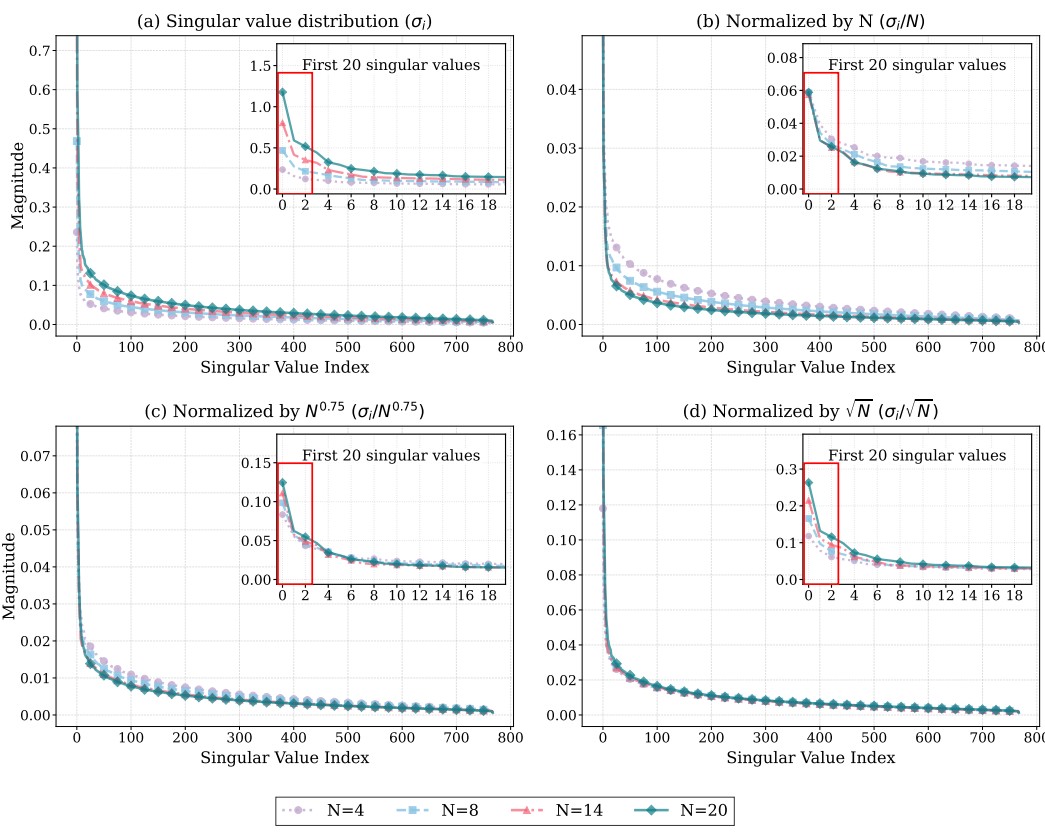

Figure 10: Normalized singular value distributions for Layer 0, mlp.c_proj. (a) Regular singular value distribution. (b) Normalization by $N$ leads to the several largest singular values overlapping, indicating linear scaling ($\mathcal{O}(N)$) typical of common components. (c) As the normalization exponent decreases to $N^{0.75}$, the largest singular values begin to separate, while the smaller singular values begin to overlap, illustrating the transition between common and unique components. (d) Finally, normalizing by $\sqrt{N}$ causes the smaller singular values (after the 10th singular value) to overlap completely, indicating that the tail scales at $\mathcal{O}(\sqrt{N})$.

To validate this, we analyze the singular value distributions of the task vectors normalized by different scaling factors $N^\gamma$ for the MLP projection layer. If a set of singular values scales as $\mathcal{O}(N^\gamma)$, dividing them by $N^\gamma$ should cause the distributions for varying $N$ (where $N \in \{4, 8, 14, 20\}$) to collapse onto a single curve. Conversely, if the scaling law is incorrect for a specific component, the curves will diverge.

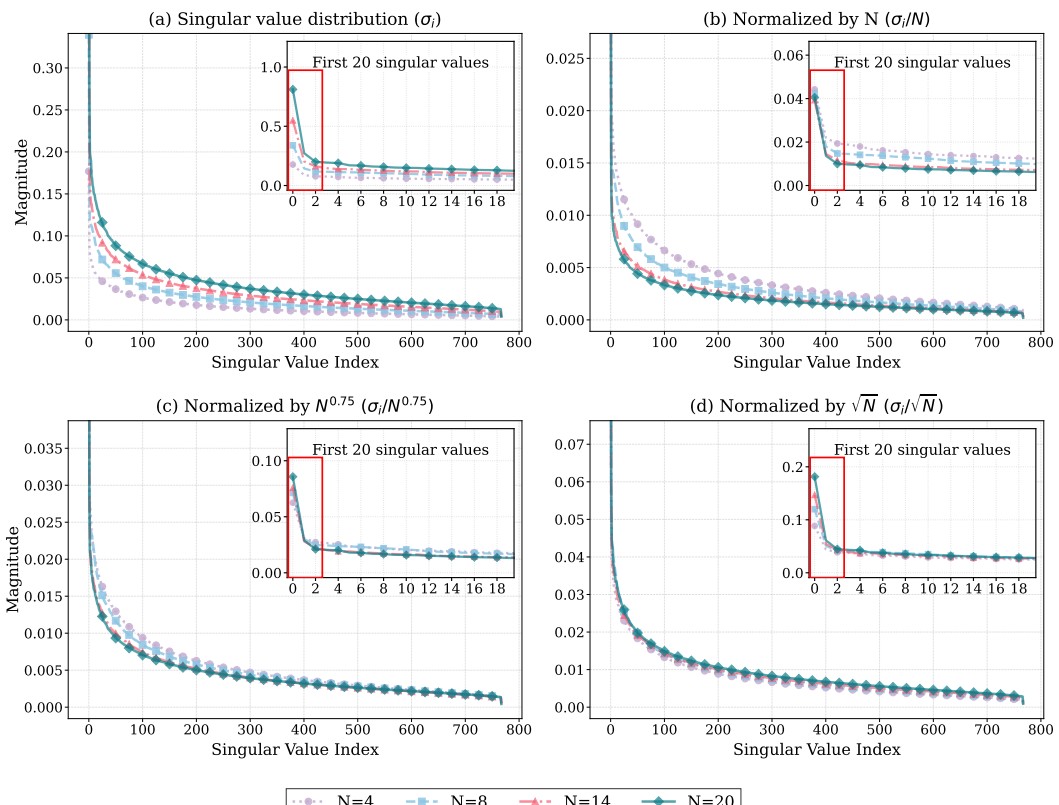

Figure 11: Normalized singular value distributions for Layer 5, mlp.c_proj. (a) Regular singular value distribution. (b) Normalization by $N$ leads to the several largest singular values overlapping, indicating linear scaling ($\mathcal{O}(N)$) typical of common components. (c) As the normalization exponent decreases to $N^{0.75}$, the largest singular values begin to separate, while the smaller singular values begin to overlap, illustrating the transition between common and unique components. (d) Finally, normalizing by $\sqrt{N}$ causes the smaller singular values to overlap completely, indicating that the tail scales at $\mathcal{O}(\sqrt{N})$. The divergence between linear scaling (common components) and square-root scaling (unique components) remains consistent for intermediate layers.

Figures 10 through 12 visualize this analysis for layers of varying depths for the second MLP layer (MLP projection layer). We observe two distinct regimes that consistently validate our theoretical bounds:

1. **Common Regime:** In panel (b) of all figures, normalizing by the number of tasks $N$ causes the several largest singular values to overlap almost perfectly. This alignment confirms that the dominant spectral components scale linearly, validating that they represent common information shared across tasks that accumulates constructively.

2. **Unique (Task-Specific) Regime:** In contrast, the smaller singular values do not overlap when normalized by $N$. However, as shown in panel (d), normalizing by $\sqrt{N}$ causes these smaller singular values to collapse onto a single curve. This confirms that the task-specific information behaves spectrally like orthogonal noise, scaling at $\mathcal{O}(\sqrt{N})$.

Panel (c), normalized by the intermediate factor $N^{0.75}$, illustrates the complexity of the spectral transition. While the separation between regimes is clearly visible, we observe that the scaling behavior

of the largest singular values is not always uniform; in some instances, dominant components may exhibit tighter overlap under this intermediate normalization factor than under the strict linear ($N$) normalization. This suggests that commonality is likely a spectrum rather than a binary property, where certain features may be shared across varying subsets of tasks, resulting in effective scaling rates that deviate slightly from the theoretical $\mathcal{O}(N)$ ideal. Nevertheless, panel (d) clearly visualizes that the largest singular values do not overlap for $\mathcal{O}(\sqrt{N})$, indicating that they scale at a larger rate than the smaller singular values.

Furthermore, this trend is robust across network depth. Figures 11 and 12 demonstrate that this spectral structure is not an artifact of early layers but a fundamental property of the weight space. In deeper layers (Layer 10), the separation between the $\mathcal{O}(N)$ common and $\mathcal{O}(\sqrt{N})$ unique components becomes even more pronounced. These results not only validate our theoretical assumptions in a real-world setting, but also confirm that rank collapse is an intrinsic consequence of the dominance of the common over task-specific information fundamental to Task Arithmetic.

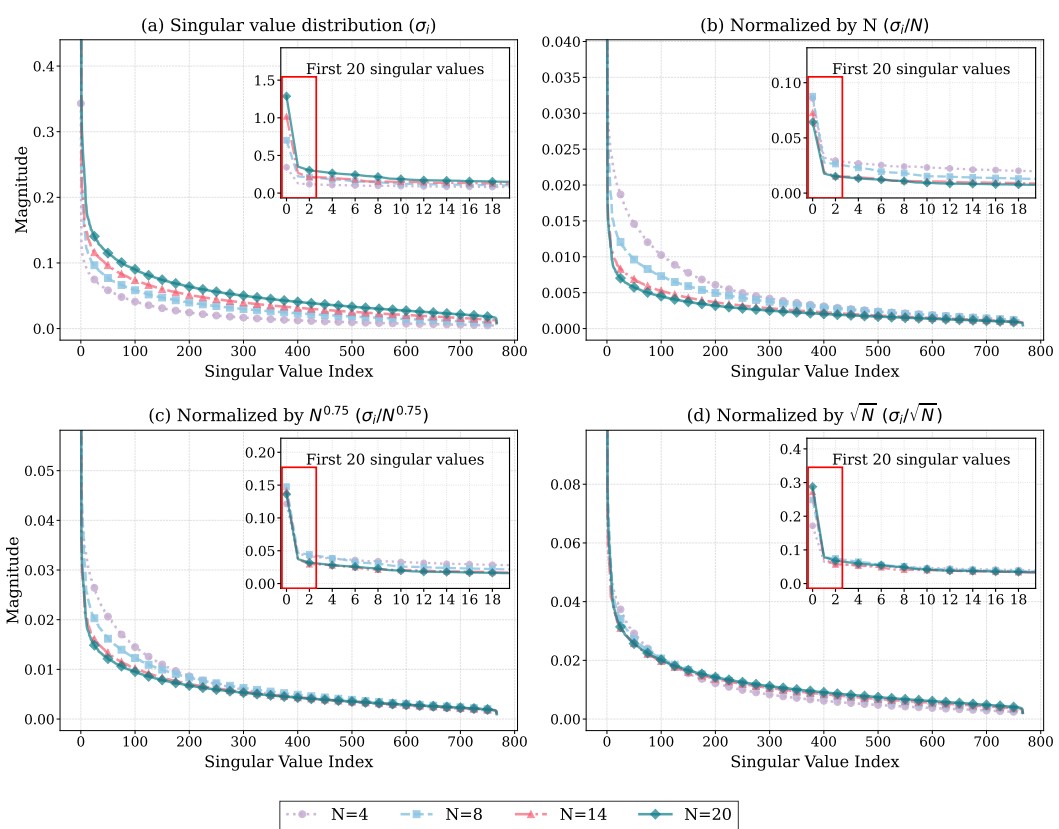

Figure 12: Normalized singular value distributions for Layer 10, mlp.c_proj. (a) Regular singular value distribution. (b) Normalization by $N$ leads to the several largest singular values overlapping, indicating linear scaling ($\mathcal{O}(N)$) typical of common components. (c) As the normalization exponent decreases to $N^{0.75}$, the largest singular values begin to separate, while the smaller singular values begin to overlap, illustrating the transition between common and unique components. (d) Finally, normalizing by $\sqrt{N}$ causes the smaller singular values to overlap completely, indicating that the tail scales at $\mathcal{O}(\sqrt{N})$. In deeper layers, the separation between the common and unique scaling regimes becomes even more pronounced, indicating that a large majority of the singular values correspond to task-specific information.

# F  ADDITIONAL ABLATION RESULTS

We perform various ablation experiments for the boosting threshold $\beta$ across different settings. In Table 7, we report the effect of the threshold for Task Arithmetic + Subspace Boosting. For the

simple case of $0.0$, the performance is consistently the best. Therefore, we advise the practitioner to set the threshold to $0.0$ by default.

Table 7: Boosting threshold ($\beta$) ablation for Task Arithmetic + Subspace Boosting

| Threshold | ViT-B/32 | | | ViT-B/16 | | | ViT-L/14 | | |
|---|---|---|---|---|---|---|---|---|---|
| | 8 Tasks | 14 Tasks | 20 Tasks | 8 Tasks | 14 Tasks | 20 Tasks | 8 Tasks | 14 Tasks | 20 Tasks |
| $\beta = 0.00$ | **83.1** | **75.8** | 63.7 | **87.7** | **82.0** | 71.3 | **91.4** | **86.2** | **80.6** |
| $\beta = 0.01$ | 82.6 | 75.3 | 63.2 | 87.4 | 81.7 | 70.9 | 90.8 | 85.8 | 80.1 |
| $\beta = 0.02$ | 82.0 | 75.2 | **66.4** | 87.2 | 80.3 | **71.6** | 86.2 | 83.8 | 79.3 |

Table 8 shows the performance for different boosting thresholds $\beta$ for Task Arithmetic + Subspace Boosting + LiNeS, depending on the layer type. For both the fully-connected layers as well as the attention weight layer, we observe that the best performance is often achieved when setting both thresholds to $0.00$. This shows that $\beta$ is quite robust to tuning and in all of our other results, we rely on one shared $\beta$ for both layer types.

Table 8: Ablation of optimal $\beta$ depending on layer.

| FC Layers | Attn Layer | 8 Tasks | 14 Tasks | 20 Tasks |
|---|---|---|---|---|
| 0.00 | 0.00 | **87.7** | **82.0** | 71.3 |
| 0.00 | 0.01 | 87.6 | 82.0 | 71.2 |
| 0.01 | 0.00 | 87.6 | 81.9 | 71.2 |
| 0.01 | 0.01 | 87.4 | 80.9 | 71.0 |
| 0.02 | 0.00 | 87.4 | 80.9 | **71.8** |
| 0.02 | 0.01 | 87.4 | 80.6 | 71.5 |

# G   HO-GSVD: ADDITIONAL INFORMATION & DETAILS

---

**Algorithm 2:** HO-GSVD with Subspace Boosting

---

**Input:** $\Delta_m = \{\Delta_{m_1}, \ldots, \Delta_{m_N}\}$ – the individual task vectors for each model.
**Output:** $\Delta_{boost} = \{\Delta_{boost_1}, \ldots, \Delta_{boost_K}\}$ – the *updated* merged task vectors for each component in the model.

1   $\Delta_{boost} \leftarrow \emptyset$;          ◁ initialize the task vector for merging.
2   $S_\pi \leftarrow 0$;          ◁ initialize $S_\pi$ to zero for all pairwise entries.
3   **foreach** $\Delta_{c_{k,i}} \in \Delta_{m_i}$ **do**
4      **foreach** $\Delta_{c_{k,j}} \in \Delta_{m_j}$ **do**
5          $S_\pi(i,j) \leftarrow$ calculateS$(\Delta_{c_{k,i}}{}^T \Delta_{c_{k,i}}, \Delta_{c_{k,j}}{}^T \Delta_{c_{k,j}})$;    ◁ calculate $S_\pi$ for each pair of components $k$.
6      $(\Lambda, V_{\text{shared}}) \leftarrow \text{eig}(S_\pi)$          ◁ compute the eigendecomposition of $S_\pi$.
7      **foreach** $i \in N$ **do**
8          $(U_i, \Sigma_i) \leftarrow$ calculate\_sing\_matrices$(S_\pi, \Delta_{c_{k,i}})$   ◁ calculate left singular vector matrix and the singular value matrix.
9      $U_{\text{ortho}} \leftarrow$ calculate\_ortho\_U$(U_1, ..., U_N)$    ◁ calculate the orthonormal matrix for all Us via Generalized Procrustes.
10      $V_{\text{ortho}} \leftarrow$ calculate\_ortho\_V$(V_{\text{shared}})$    ◁ calculate orthonormal matrix for V via Procrustes.
11      $\Sigma_{\text{sum}} \leftarrow sum(\Sigma_1, ..., \Sigma_N)$    ◁ sum up the singular values into one shared matrix.
12      $\Sigma_{\text{boosted}} \leftarrow$ subspace\_boosting$(\Sigma_{\text{sum}})$    ◁ boost the singular values with subspace boosting.
13      $\Delta_{boost_k} \leftarrow U_{\text{ortho}} \cdot \Sigma_{\text{boosted}} \cdot V_{\text{ortho}}^T$;    ◁ recompute the task vector using new boosted singular values.
14      $\Delta_{boost} \leftarrow \Delta_{boost} \cup \Delta_{boost_k}$;          ◁ add the updated component.
15   **return** $\Delta_{boost}$          ◁ return merged task vector.

---

### G.1 HO-GSVD DETAILS AND HIGHER-ORDER SUBSPACE BOOSTING

Given a set of $N$ matrices $A_1, \ldots, A_N$, HO-GSVD decomposes each matrix $A_i = U_i \Sigma_i V^T$, $i = 1, \ldots, N$ resulting in *distinct* $U_i \in \mathbb{R}^{m_i \times n}$, $\Sigma_i \in \mathbb{R}^{n \times n}$ and a *shared* subspace $V \in \mathbb{R}^{n \times n}$, identical for all factorizations. To obtain $V$, we solve the eigensystem $S_\pi V = V \Lambda$ of the arithmetic mean $S_\pi$ of all pairwise quotients $D_{i,\pi} D_{j,\pi}^{-1}$:

$$S_\pi = \frac{1}{N(N-1)} \sum_{i=1}^{N} \sum_{j=i+1}^{N} (D_{i,\pi} D_{j,\pi}^{-1} + D_{j,\pi} D_{i,\pi}^{-1}), \tag{31}$$

with $D_{i,\pi}$ defined as:

$$D_{i,\pi} = A_i^T A_i + \pi A^T A, \pi \geq 0, \tag{32}$$

where $A = [A_1^T, \ldots, A_N^T]^T$. Following Kempf et al. (2023), we add the previous regularization term $A^T A$ and scale it by $\pi$ to extend HO-GSVD's applicability to *rank-deficient matrices*. This phenomenon frequently occurs for task vectors, as particularly visualized for merged task vectors in Fig. 1a. We set $\pi$ to $10^{-2}$ by default. As previously described, HO-GSVD introduces a shared subspace *identical across all inputs*. This enables the *direct comparison of different models* and the identification of common or unique subspaces for different tasks. HO-GSVD allows us to perform merging in a more interpretable manner by operating compositions over shared subspaces.

However, unlike standard SVD, the left and right singular matrices are generally not orthonormal (Ponnapalli et al., 2011; Kempf et al., 2023). In order to perform subspace boosting with this method, we must first orthonormalize the respective matrices. This is achieved by solving the Generalized Orthogonal Procrustes problem (Golub & Van Loan, 2013; Schönemann, 1966). Afterwards, the weight matrices can be reconstructed using these newly orthonormal matrices while preserving a shared subspace across all models. Please refer to Algorithm 2 for more details.

### G.2 HIGHER-ORDER SUBSPACE BOOSTING MERGING PERFORMANCE

Similarly to its standard SVD counterpart *Subspace Boosting*, HO-GSVD can also be used for highly performant model merging, using Algorithm 2, which we refer to as *Higher-Order Subspace Boosting*. Table 9 showcases the performance of Higher-Order Subspace Boosting + LiNeS against other performant methods. We observe that the method consistently achieves strong performance. For ViT-B/32, for 8 tasks, it is the second best method, whereas for 14 tasks, it achieves the same performance as our best-performing Task Arithmetic variant. This showcases that *Higher-Order Subspace Boosting* is a strong model merging method and a potentially suitable plug-in variant for standard *Subspace Boosting*, but with the additional benefit of operating over shared composition spaces for improved interpretability. Similar results can be seen for ViT-B/16, establishing *Higher-Order Subspace Boosting* as a strong, but also interpretable model merging method.

Table 9: Higher-Order Subspace Boosting (Higher-Order SB) Performance compared against *Subspace Boosting* variants utilizing Task Arithmetic (TA), TIES, and Consensus Merging (CSM).

| Method | ViT-B/32 | | | ViT-B/16 | | |
|---|---|---|---|---|---|---|
| | 8 Tasks | 14 Tasks | 20 Tasks | 8 Tasks | 14 Tasks | 20 Tasks |
| TA + SB + LiNeS | **85.6** | **80.8** | 77.2 | **88.8** | **84.7** | 80.0 |
| TIES + SB + LiNeS | 83.8 | 79.1 | 75.9 | 87.4 | 83.3 | 79.7 |
| CSM + SB + LiNeS | 84.4 | 80.3 | **77.2** | 87.6 | 84.2 | **80.0** |
| Higher-Order SB + LiNeS | 84.5 | 79.8 | 75.6 | 88.5 | **84.7** | 79.1 |

## H ADDITIONAL EXPERIMENTS LEVERAGING HO-GSVD

Leveraging HO-GSVD, we can revisit again in more detail the Alignment Matrices introduced in the main paper. Figure 13 shows the resulting Alignment Matrix for the attention block's projection layer across different layers. Subfigure 13d shows the mean Alignment Matrix across all layers and components. While for layer 0, the values are consistently low and showing high overlap

between models, for the deeper layers, the alignment scores start becoming more defined. This confirms that shallow layers learn general features that are shared across multiple models (Wang et al., 2024a), hence the high overlap of important dimensions, whereas the later layers showcase more task-specific information and separation into subspaces.

Table 10: Model selection comparison for 8 experts evaluated on all 20 tasks.

| Method | Accuracy | Normalized Accuracy |
|---|---|---|
| Random selection | $67.5 \pm 2.3$ | 72.5 |
| HO-GSVD selection | **69.7** | **75.2** |

We perform an additional experiment by merging 8 models out of 20 chosen either randomly or via our optimal selection approach. Random selection is performed 20 times, and we report the average. The results are shown in Table 10. The performance of the merged model is evaluated across all 20 tasks (including those not involved in the merging process). When merging via optimal model selection, the method accurately chooses a subset of models that performs better than average. Based on both experiments, one can observe that optimal model selection outperforms average random model selection by several percent, highlighting the effectiveness of using HO-GSVD to select optimal models for merging.

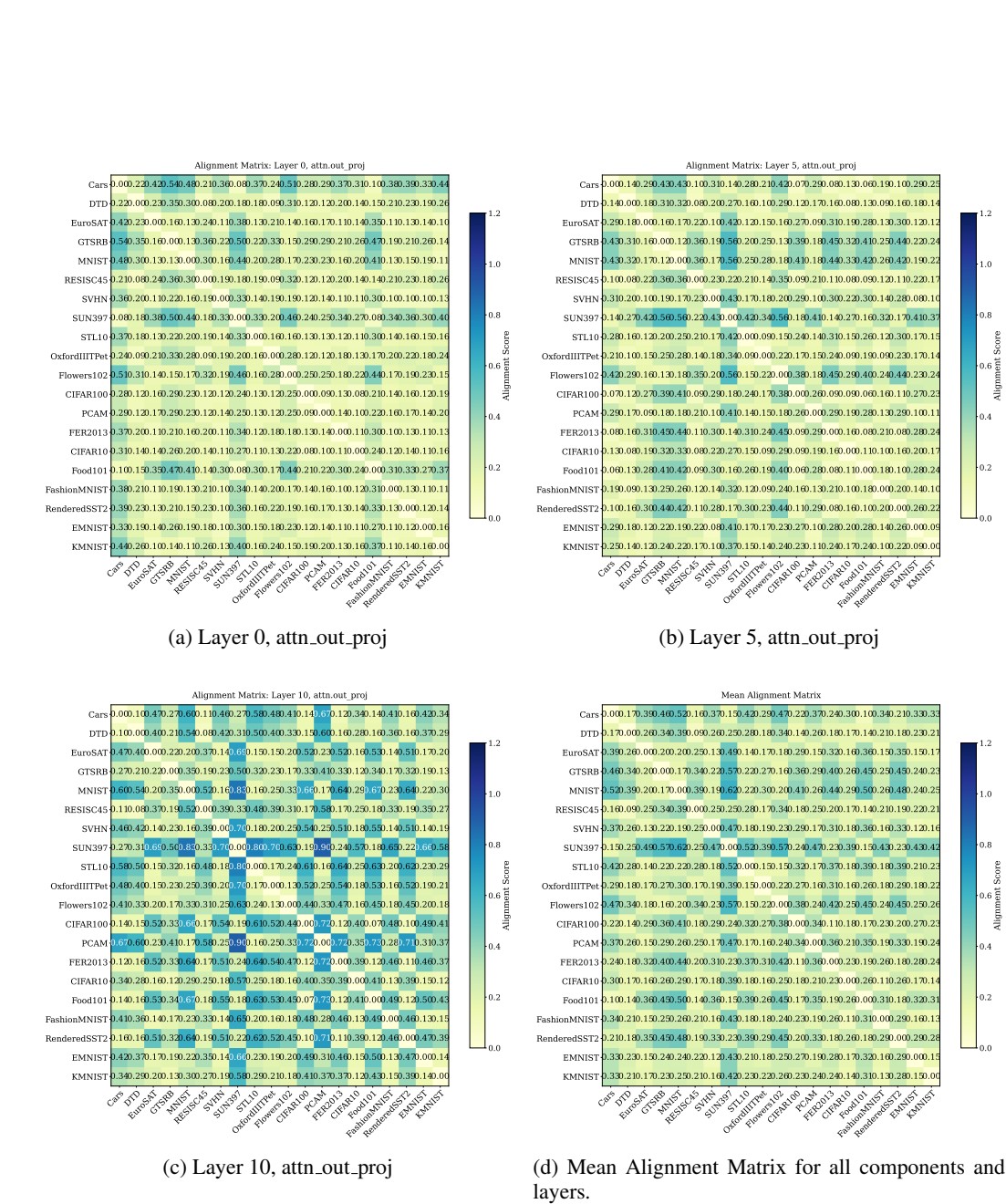

(a) Layer 0, attn_out_proj

(b) Layer 5, attn_out_proj

(c) Layer 10, attn_out_proj

(d) Mean Alignment Matrix for all components and layers.

Figure 13: Alignment matrices for attn_out_proj for different layers as well as the mean Alignment Matrix for all components.

