# OpenReview forum: "Subspace-Boosted Model Merging"
_ICLR.cc/2026/Conference — Submitted to ICLR 2026_

### Official Review · Reviewer_RgdQ · 2025-10-25

**Soundness:** 3
**Presentation:** 3
**Contribution:** 2
**Rating:** 4
**Confidence:** 5

**Summary:**

The work tackles task-vector-based model merging, where the differences between task-specific finetunings and their base models (task vectors) are aggregated to obtain one single multi-task model that can perform all the tasks at once. In particolar, the paper investigates the problem of rank collapse when merging many tasks: the more tasks are added, the higher the singular values for the most important directions, contracting the spaces into a few dominant directions. The work proposes clamping the singular values to avoid some of the singular directions overshadowing the others. It also proposes higher-order SVD to obtain a space that is shared across the tasks, making different task-specific weights more comparable in the latter. The method is tested on ViT-B/32, ViT-B/16 and ViT-L/14 on the common 8-, 14, and 20-tasks benchmarks.

**Strengths:**

- The method is interesting and intuitive. Current SVD-based methods either do not consider any common subspace (TSV [1]) or obtain one just by summing the task vectors before the SVD takes place (Iso-C [2]). Producing this one through Higher-Order SVD seems principled and effective. The method is applicable to any existing task-arithmetic variant, including e.g. TIES and Consensus Merging. It is also more efficient than comparable SVD-based methods, which is by itself not a big thing in standard model merging but could be useful in applications requiring iterative merging (e.g. federated learning). Being data- and tuning-free, the method is broadly applicable and in line with the considered literature.
- The paper is well written and easy to read, with clear figures that are immediate to grasp. The formalization is intuitive and the paper structure is well-thought.
- The results are competitive with the state-of-the-art, and the experimental evidence is extensive and considers all the relevant baselines, model architectures and benchmarks.

**Weaknesses:**

- The evidence for the main motivation, i.e. the rank collapse, is somewhat limited: in layer 10, one of the most significant, the trend is actually the opposite: N14 and N20 have higher stable rank than N8. This suggests that it has more to do with the particular composition of the merging set rather than the number of tasks alone. For a similar analysis to be performed, one should try to average over subsets of increasing cardinality, possibly many of them so to rule out the difference in composition.
- The novelty is somewhat limited when compared to Iso-C [1]. The main contribution seems to be solving the rank collapse phenomenon, but this seems to be already tackled by Iso-C/Iso-CTS [1], although in a simpler manner. Simplicity however has its benefits, as [1] does not require a beta hyperparameter.
- From the performance standpoint, the approach only outperforms the current state-of-the-art in the ViT-B-32 setting, while remaining below Iso-CTS for larger architectures.
- The alignment method is not well motivated and discussed. I don’t fully understand what’s the intuition of e.g. MNIST and EuroSAT being most aligned to SUN397. How does this alignment measure correlate with the interference measures proposed in TSV and Iso-C? The subspace alignment studied in Iso-C would be particularly interesting as from my understanding it also measures similarity in the singular vector space, although differently.
- The whole “interpretable merging” point seems fairly oversold. It is not immediately clear to me what added interpretability stems from the proposed alignment matrices. I might have missed the point, and I would be happy to be convinced otherwise on this aspect.

Given the current strengths and weaknesses, I am inclined to reject the paper: from the performance standpoint, it does not distance itself by comparable baselines that are also similarly motivated, and the remaining benefits (e.g., interpretability) are not properly explored and discussed.

[1 Marczak, Daniel, et al. "No Task Left Behind: Isotropic Model Merging with Common and Task-Specific Subspaces." ICML 2025.

[2] Gargiulo, Antonio Andrea, et al. "Task singular vectors: Reducing task interference in model merging." *Proceedings of the Computer Vision and Pattern Recognition Conference*. 2025.

**Questions:**

- What is the proper way to look at the alignment matrix? what are its immediate implications? how does it correlate with other existing measures? (see weakness 4).
- Figure 3 does not specify the architecture, the task nor the layer.
- Why is A used in place of Delta (common choice in previous literature?) This seems like a peculiar choice, given that the alignment matrix is termed **A**.
- I don’t fully understand the I_{>1} notation. I get that it refers to the common components, but why is it expressed in this way?
- M is also somewhat confusing as might lead to think of a matrix instead of a scalar. Also some papers use it for the merged model.
- What sort of interpretability can we derive from the alignment matrices?

---

> ### Author Response · Authors · 2025-11-23
> **Reply to Reviewer RgdQ part 1**
>
> We thank the reviewer for the comprehensive review of our submission and the constructive feedback.
>
> Following the reviewer’s comments as well as the comments of other fellow reviewers, we have made significant updates to the revision. **We now include a completely new theoretical subsection 3.1 that proves that rank collapse is inevitable under mild assumptions.** We also provide theoretical justification for Subspace Boosting that we hope significantly improves the novelty and rigor of our work.
>
> We address reviewer's concerns in detail below.
>
> ---
> **The evidence of rank collapse is limited**
>
> Following the reviewer’s comment that the evidence for rank collapse is limited, we now provide new Propositions 1-3, providing significantly enhanced theoretical grounding for our previous empirical evidence of rank collapse.
>
> _Proposition 3 states that as more models are merged, the common information of the merged task vector will dominate the task-specific information. For the simple case of averaging, the stable rank will collapse to the stable rank of the common component, proven in Proposition 2._
>
> While the reviewer is correct that taking different subsets of models would likely result in slightly different plots for rank collapse, we irrefutably prove in the previous propositions that as more models are merged, the rank will inevitably collapse. The layer 10 fluctuation observed by the reviewer is likely due to a specific subset composition, as the reviewer suspected. **However, Proposition 3 proves that as more models are merged, the common component will dominate the task specific information and lead to rank collapse.** We hope this addresses the reviewer’s concern about the evidence for rank collapse.
>
> ---
>
> **Limited novelty compared to  Iso-C/Iso-CTS**
>
> While we agree that Iso-C/Iso-CTS [1] inadvertently alleviates rank collapse, our novelty comes from different angles. Unlike Iso-c/Iso-CTS [1], **we are the first to discover, quantify and prove that rank collapse is inevitable for Task Arithmetic-based methods.** We also propose the first interpretable model merging framework based on HOG-SVD.
>
> We discover, quantify and empirically evaluate rank collapse, which the reviewer agrees is extensive and considers all relevant baselines, architectures and benchmarks. Furthermore, whereas Iso-C is only applied to one method, we apply our method to a set of popular Task Arithmetic-based methods such as TA, TIES and TALL Masks, which shows that our method is general for the whole class of Task Arithmetic-based methods. We believe that our performance improvement for a range of Task Arithmetic-based methods is significantly more unexpected, novel and useful for the overall community.
>
> Moreover, our new theoretical results prove that rank collapse is inevitable under the mild condition of near orthogonality of task vectors, which is empirically validated by Ilharco et al. [1]. Hence, **we are the first to diagnose and prove why merging an increasing number of models will inevitably lead to lower performance**. By adding these theoretical results, we believe we have greatly enhanced the soundness and novelty of our results.
> The above results are all novel to our work and have not been discussed by the authors of Iso-c/Iso-CTS [1].
>
> **Subspace Boosting efficiency**
>
> Finally, our method is significantly more efficient than competing methods for the following reasons: TSV-M performs SVD on every matrix individually to get Task Singular Vectors. In essence, this is O(N), where N is the number of tasks for merging and afterwards once more for the decorrelation (O(1)). On the other hand, Subspace Boosting only applies one SVD on the merged matrix, regardless of the number of tasks, which is O(1). Hence, TSV-M scales linearly whereas Subspace Boosting scales in O(1), making our method significantly more efficient than TSV-M and subsequently, Iso-CTS (because  Iso-CTS is an extension on top of TSV-M) . This efficiency is especially important for merging larger models such as multi-billion parameter models.
>
> **Subspace Boosting $\beta$ hyperparameter**
>
> Regarding the hyperparameter $\beta$, we provide the reader with a simple heuristic to set it to 0 by default, if no hyperparameter tuning is desired. Therefore, our method can also be used without a hyperparameter. In fact, we follow this rule ourselves, since the results in Table 2. For the Language Domain does not involve any hyperparameter tuning of $\beta$ and we set it to 0 by default, improving the TA+LiNeS results from 67.6 &rarr; 75.3 obtained by Subspace Boosting with default  $\beta=0.0$.

---

> > ### Author Response · Authors · 2025-11-23
> > **Reply to Reviewer RgdQ part 2**
> >
> > **Results in terms of State-of-the-art**
> >
> > The main goal of our work is to investigate why merging performance declines as more models are merged. Initially, we did not aim to reach State-of-the-art results, however, surprisingly, we do achieve them. In Table 4, **our method outperforms Iso-CTS for 4/9 tasks, in essence, showcasing that our method is comparable to Iso-CTS**. Moreover, the actual performance difference between the two methods is marginal and is **less than 0.5% for 6/9 tasks**. Finally, the largest performance difference is actually in our advantage for the merged ViT-B/32 model for 8 tasks, with a **performance improvement of 1.8% for Subspace Boosting compared to Iso-CTS.**
> >
> > In fact, we believe that reaching SOTA results with two distinct approaches is valuable in itself for the community, since the community can understand what leads to superior merging performance better.
> >
> > ---
> > **Alignment method motivation**
> >
> > We thank the reviewer for highlighting that the HO-GSVD section could be improved. We have improved the introduction of Section 4.2 to better motivate why HO-GSVD is necessary and what limitations of standard Subspace Boosting it solves.
> >
> > We define 'Alignment' as non-interference. High alignment scores indicate tasks occupy different subspaces and thus merge easily. In essence,  to answer the reviewer’s question about MNIST and EuroSAT being most aligned to SUN397, this means that MNIST, EuroSAT and SUN397 all have vastly different information (since MNIST contains digits, EuroSAT - satellite imagery and SUN397 - scene understanding, these three tasks require different information in order to classify these samples).  Therefore, they are more suitable for merging (High alignment) and will interfere less in the corresponding subspaces.
> >
> > To further illustrate this, In Fig. 10. d. the task vectors for MNIST, EMNIST, and KMNIST (digits, latin characters, japanese characters) have relatively low alignment (are very similar, since they probably rely on the same patterns to recognise the digits and characters) of around 0.24. This means that merging these tasks will be more difficult since they rely on the same subspaces, hence causing high interference and low alignment (mergeability). This also can be empirically validated, since MNIST, EMNIST, and KMNIST are all low-resolution black-and-white datasets, thus the models likely extract similar features and operate in a similar manner, which would lead to their parameter space being more similar.
> >
> > While the Subspace Alignment Ratio introduced in [1] is a valid metric to evaluate whether the merging succeeded,  we believe that the metric is naive and not granular enough for interpretability purposes.
> >
> > First, the method performs SVD on the merged matrix. This approach does not separate the contribution of each individual task or interference between tasks per dimension. There is no way to disentangle the task-specific information per dimension. Hence, it is more suitable as a metric to optimize, rather than to deeply interpret model merging.
> >
> > **Comparison to Subspace Alignment Ratio**
> >
> > On the other hand, we provide a more mathematically rigorous way to evaluate task similarity using the Alignment matrix. By using HO-GSVD, we utilize one shared basis V across all tasks. This allows for direct comparison of how different tasks are utilizing the same dimension. This type of granularity cannot be achieved with the Subspace Alignment Ratio. This also allows one to evaluate not only how much models interfere with each other, but how they share the underlying parameter space (common vs unique components).
> >
> > Moreover, unlike the Subspace Alignment Ratio (SAR), our approach allows us to explicitly disentangle the signal by categorizing “common” and “unique” subspaces. This provides us the ability to distinguish whether a dimension is crucial for multiple tasks (high interference) or unique to one (high alignment), providing a better interpretation of task similarity.
> >
> > Finally, unlike SAR, we show that our Alignment matrix effectively evaluates the interference by successfully selecting a subset of models that merge well, outperforming random baselines in Tables 5 and 10, thus also providing actionable interpretability.

---

> > > ### Author Response · Authors · 2025-11-23
> > > **Reply to Reviewer RgdQ part 3**
> > >
> > > **Interpretability motivation**
> > >
> > > We would like to point out that interpretable merging is a secondary result of our research.
> > > Nevertheless, as mentioned above., utilizing our HO-GSVD approach provides granular interpretability per dimension. It allows one to disentangle each dimension and task and understand which dimension is highly conflicting and what tasks are highly interfering with each other, in other words, not well-aligned. In other words, each dimension can be inspected, and the common vs unique subspaces can be quantified and distinguished.
> > >
> > > Moreover, we see that the Alignment matrix confirms insights such as that earlier layers learn fine-grained information that is common among tasks whereas later layers learn task-specific information. This can be seen in Fig.10. Since the alignment scores are low for layer 0, this means that the tasks interfere more with each other, whereas for layer 10, we see that the alignment is much higher, indicating that the tasks now live in specific subspaces.
> > >
> > > ---
> > >
> > > **Missing architecture in the figure.**
> > >
> > > We apologize for this oversight for Figure. 3, the architecture is ViT-B/16 (mentioned in the preceding paragraph). The singular values are plotted for layer 5, attention out projection layer.
> > >
> > > ---
> > >
> > > **HO-GSVD notation**
> > >
> > > We utilize the same notation as used in the original paper [4] to be consistent with the existing literature for HO-GSVD, so that the reader can reference and compare to the original works.
> > >
> > >
> > >
> > >
> > > In conclusion, we hope that our significant new theoretical results as well as our answers to the reviewer’s questions convince the reviewer of the enhanced novelty and rigour of our work.
> > >
> > >
> > > ---
> > >
> > > [1] "No Task Left Behind: Isotropic Model Merging with Common and Task-Specific Subspaces." Marczak, et al. ICML 2025.
> > >
> > > [2] Editing Models with Task Arithmetic. Ilharco et al. https://arxiv.org/abs/2212.04089.
> > >
> > > [3]  "Task singular vectors: Reducing task interference in model merging." Gargiulo et al. Proceedings of the Computer Vision and Pattern Recognition Conference. 2025.
> > >
> > > [4] “A Higher-Order Generalized Singular Value Decomposition for Rank Deficient Matrices”. Kempf et al. 2022. https://arxiv.org/abs/2102.09822

---

### Official Review · Reviewer_Eqcf · 2025-10-31

**Soundness:** 2
**Presentation:** 3
**Contribution:** 2
**Rating:** 4
**Confidence:** 3

**Summary:**

This paper identifies "rank collapse" in the task vector space as a fundamental limitation in model merging, which explains why performance gains diminish as more expert models are combined. To solve this, the authors propose Subspace Boosting, a training-free method that uses Singular Value Decomposition (SVD) to decompose task vectors and explicitly enhance underutilized dimensions, thereby maintaining the effective rank and significantly improving merging efficacy by over 10% on vision and language tasks. Additionally, the paper introduces the use of Higher-Order Generalized SVD (HO-GSVD) as a novel framework to quantify task similarity, offering a new interpretable perspective on model merging and enabling principled expert selection.

**Strengths:**

- Unlike prior works that only observed diminishing performance with more merged experts, this study provides a mechanistic explanation from a task vector space perspective: as more experts are merged, task vectors suffer from rank collapse.

- The proposed Subspace Boosting addresses rank collapse in a highly practical manner. Operating via singular value decomposition (SVD) on merged task vectors, it boosts underutilized small singular values to maintain effective rank.

**Weaknesses:**

- The third part quantifies Rank collapse only relying on "Stable Rank" and "Cumulative Energy Rank" (for example, Formula 2 defines stable rank as the ratio of the sum of squares of singular values to the square of the maximum singular value). However, the universality of these two indicators for the "correlation degree of model fusion performance" has not been fully demonstrated. The manuscript only demonstrates the negative correlation between the stable rank and performance through experiments of the ViT-B/16 model (Figures 2 and 3), but does not verify:
     - a . In different model architectures (such as the language model T5), whether the stable rank can still effectively reflect the impact of rank collapse on performance - for instance, the dimension of the weight matrix and the layer structure of T5 are significantly different from those of ViT, the distribution rules of singular values may be different, and the "effective rank" representation ability of the stable rank may fail;
     - b. In extreme scenarios (such as fusing two models or fusing models with highly similar tasks), will there be a counterexample of "low rank but high performance" in the stable rank? If so, it indicates that this metric cannot be used alone as a criterion for determining rank collapse.

- Subspace enhancement relies on the hyperparameter β (lift threshold) to determine the singular value cutoff point to be enhanced. However, this paper only found through experiments that the performance is stable when β∈{0,0.01,0.02} (Table 3a), without providing a theoretical explanation when the task type changes, does the optimal value range of β remain stable? If β needs to be re-tuned according to the scene, the practicality of the method will significantly decline, but the documentation has not verified this boundary condition.

- The experiment only selects "same mode, same type" tasks.  All visual tasks are classification tasks and do not include non-classification tasks such as detection and segmentation. All language tasks are QA and NLP classification tasks (such as sentiment analysis), and do not include generation or translation tasks. The "cross-type task fusion" scenario - such as fusing classification and detection tasks - has not been verified to see if subspace enhancement can still improve performance. If the task vector conflicts of cross-type tasks are more significant, the method's effectiveness may drop significantly, and the generalization of existing conclusions is limited.

**Questions:**

- Whether Subspace Boosting can simultaneously improve performance on both visual and language sub-tasks?

---

> ### Author Response · Authors · 2025-11-23
> **Reply to Reviewer Eqcf**
>
> We thank the reviewer for their thoughtful feedback, especially regarding improving the generalizability of our results to different architectures and task types.
>
> Following this reviewer’s comments as well as the comments of fellow reviewers, we significantly enhance our work in the new revision. **We add new theoretical results in subsection 3.1, theoretically justifying both rank collapse and our chosen metrics such as the stable rank.** These theoretical results are agnostic of the task type or architecture and only rely on the fact that task vectors are nearly orthogonal.
>
> Following these theoretical results, we discuss the mentioned points below:
>
> ---
> **Rank Collapse is inevitable in Task-Arithmetic-based models**
>
> In the new revision, Proposition 1. and 3. prove that rank collapse is inevitable as more models are merged, regardless of the architecture or task type (this is also empirically validated in the paper with ViTs in Table 1 and T5s in Table 2). This holds under the mild assumption that tasks are nearly orthogonal, which is shown in  [A]. In essence, rank collapse is inevitable as more models are merged, since the common information among tasks will dominate the task-specific information at a rate of $O(\sqrt{N})$, for N models.
>
> ---
>
> **Stable rank reflects rank collapse**
>
> Following Proposition 1, we prove in Proposition 2. that the stable rank will converge to the stable rank of the common component as more models are averaged. Therefore, this shows that the stable rank is an accurate metric for evaluating rank collapse.
>
> We acknowledge that transient cases, such as the reviewer's example of , “low rank, but high performance” for only a few merged models, can occur. However, our theoretical result holds in the large-scale limit (many averaged models).  As empirically demonstrated through our results on vision and language domains, with 2 different architectures (ViTs and T5s, in Table 1 and 2), the stable rank score accurately reflected the rank collapse of the merged task vectors.
>
> Nevertheless, our goal  is to evaluate why model merging performs increasingly poorly as more models are merged, i.e., for large N. Under this condition, the mentioned Propositions do hold and they show the inevitability of rank collapse as well as the effectiveness of the stable rank as an effective metric.
>
>
> ---
> **Robustness of $\beta$**
>
> While it is true that the $\beta$ likely changes from task type to task type, we demonstrated empirically that it is robust. In Table 3, it is validated  that the best $\beta$ is often 0.0. In fact, we mentioned the default selection of $\beta=0.0$. For example, our NLP results in Table 2,  are obtained with $\beta=0.0$, without performing hyperparameter tuning of $\beta$  (_improving the TA+LiNeS results from 67.6 &rarr; 75.3 obtained by Subspace Boosting with default  $\beta=0.0$_).
>
> We hypothesize that the optimal  $\beta$ range will be the same as long as the near orthogonality of the task vectors is similar to that observed for Vision Transformers [A]. In fact, this near orthogonality assumption seems to hold in general for other NLP tasks, such as the ones evaluated in Table 16 (MNLI for language inference, and CR for sentiment analysis) in Zeng et al.’s work [B].
>
> ---
> **Generalizability to different task and modality types**
>
> In order to generalize our results to any task and any model, we mathematically prove that rank collapse is inevitable in Propositions 1-3 under the assumption of near orthogonality of task vectors.
> Empirically, this assumption holds for various tasks, such as vision classification tasks [A], as well as various NLP tasks (NLP inference, sentiment analysis) presented in Zeng et al.’s work in Table 16 [B]. For both cases, we can see that the task vectors are near orthogonal, therefore, validating this assumption. Hence, our results generalize to different task types and architectures.
>
> Finally, we would also like to highlight that our current Vision Transformer results are already **multi-modal**, since they are based on the CLIP-based architecture, _incorporating both a text encoder and vision encoder, validating that our results already generalize to multi-modal scenarios._
>
> [A] Editing Models with Task Arithmetic. Ilharco et al. https://arxiv.org/abs/2212.04089.
>
> [B] Efficient Model Editing with Task Vector Bases: A Theoretical Framework and Scalable Approach. Zeng et al. https://arxiv.org/pdf/2502.01015v2.

---

### Official Review · Reviewer_NH2N · 2025-10-31

**Soundness:** 3
**Presentation:** 2
**Contribution:** 2
**Rating:** 2
**Confidence:** 3

**Summary:**

This paper reveals the rank collapse limitation in existing model merging methods. To address this issue, the authors propose a technique called subspace boosting, whose core idea is to boost the singular values below a certain cutoff threshold.

**Strengths:**

1. This work identifies a critical limitation in existing model merging approaches, rank collapse, and provides empirical evidence to support this finding.

2. The paper is well-organized and easy to follow.

**Weaknesses:**

1. **Potential error amplification.** Directly boosting the singular values below the cutoff point may introduce noise or bias. The authors should provide further discussion to justify the rationality of the proposed subspace boosting technique.

2. **Hyperparameter sensitivity.** The method requires manual tuning of the cutoff hyperparameter, which may limit its practicality and robustness in real-world applications.

3. **Unclear connection between HO-GSVD and rank collapse.** While HO-GSVD offers a new and interpretable perspective on model merging, it is unclear how it directly relates to the core motivation of preventing rank collapse. This conceptual gap may cause readers to lose focus on the main contribution.

**Questions:**

See weaknesses above.

**Details Of Ethics Concerns:**

No ethics concerns.

---

> ### Author Response · Authors · 2025-11-23
> **Reply to reviewer NH2N**
>
> We thank the reviewer for their feedback regarding both the strengths and weaknesses of this paper.
>
> We address your concerns in detail below.
>
> ---
> **Potential error amplification**
>
> We thank the reviewer for this insightful suggestion. We agree with the reviewer that Subspace Boosting potentially amplifies noise. **Following the remarks by several reviewers, we have revamped the paper and added substantial new theoretical results in the new Subsection 3.1**. In addition, especially relevant is our theoretical result regarding the boosting of task-specific signal vs. noise. We prove Proposition 4 in the Appendix, which states the Signal-to-noise tradeoff of Subspace Boosting.  **We prove that under mild assumptions, the task-specific information recovered by Subspace Boosting exceeds the error introduced by boosting the noise.**
>
> Therefore, we justify the effectiveness of Subspace Boosting theoretically. In other words, Subspace Boosting recovers the task-specific information which is significantly more important for performance than the detrimental effect of noise amplification.
>
>
> Empirically, we evaluated filtering the tail to potentially remove the noisy components. However, regardless of how few tail singular values were removed, the performance actually decreased. Therefore, to stick to a simple method, we just boost indiscriminately all singular values, including noisy components. This makes the method very efficient and empirically does not show performance degradation. This likely means that for a large enough number of merged models, all dimensions correspond to at least some task-specific signal.
>
> ---
>
> **Hyperparameter sensitivity**
>
> While we agree with the reviewer that the method requires the boosting threshold $\beta$, we provide empirical evidence that hyperparameter tuning of $\beta$ is robust. We show that $\beta$ is robust to tuning in Table 3. Additionally, we state in the paper that readers can simply choose $\beta=0.0$ to remove this hyperparameter tuning. In fact, for the NLP results in Table 2, we did not perform any hyperparameter tuning ourselves and set $\beta=0.0$ by default (improving the TA+LiNeS results from 67.6 &rarr;  75.3 obtained by Subspace Boosting with default  $\beta=0.0$).
>
> ---
>
> **Unclear connection between HO-GSVD and rank collapse**
>
> We thank the reviewer for pointing out that this section required improvement in clarity. We have improved the introduction of Section 4.2, which introduces HO-GSVD. We highlight the weaknesses of Subspace Boosting, which we address with HO-GSVD.
>
> To reiterate, standard SVD does not allow us to identify the task-specific dimensions. This information could be very valuable to understand which dimensions suffer from interference of different tasks, i.e., common subspaces, vs. task-specific subspaces. HO-GSVD solves this inherent limitation. This is essential, for example, for deeper evaluation and interpretation of task alignment for better model merging (Table 5, Fig.5.b.). Moreover, HO-GSVD also provides us a much better evaluation with rank collapse, since we can identify which dimensions suffer from interference or rank collapse (Fig.5.a.), without the need for external metrics such as the stable rank or cumulative energy rank.
>
> We hope we could clarify the reviewer’s concerns. In addition, following the reviewer’s first point, we hope that our significant theoretical additions to the updated revision greatly improve the overall paper and hope that the reviewer acknowledges this in their new assessment.

---

### Official Review · Reviewer_AddN · 2025-10-31

**Soundness:** 3
**Presentation:** 3
**Contribution:** 2
**Rating:** 4
**Confidence:** 4

**Summary:**

The paper introduces Subspace Boosting, a method developed to improve the effectiveness of model merging. The authors analyze the degradation in performance that occurs as the number of merged models increases, attributing it to a reduction in the effective dimensionality of the task-vector space, where variance becomes concentrated in a few dominant directions.
Subspace Boosting is implemented on top of the Higher-Order Generalized Singular Value Decomposition (HO-GSVD) framework. Additionally, the authors introduce the Alignment Matrix, which measures relationships between task vectors within the shared subspace.
Experiments on Vision Transformer (ViT-B/32, ViT-B/16, and ViT-L/14) models trained on 8, 14, and 20 tasks compare Subspace Boosting against the ISO-C and TSV baselines and show the performance obtained applying this method to some model-merging baselines ( TIES-Merging, Task Arithmetic, Consensus Merging).

**Strengths:**

- The topic is relevant to the community, addressing a problem in multi-task and model-merging research.
- The proposed Subspace Boosting method is conceptually clear and appears computationally efficient.
- The connection between singular value structure and model merging dynamics is insightful. The analysis of shared versus task-specific subspaces through the singular-value structure is particularly interesting.

**Weaknesses:**

- Some inclarities about the tables and exepriment report (see questions).
- The notion of “rank collapse” that is central in the paper could benefit from a more formal explanation.
- Algorithm 1 applies a standard SVD step. I think it is misleading to put it in the approach instead of the algorithm of the Subspace Boosting.

Minor
- In Figure 2 (a–c), the y-axis label “Value” should likely be “Stable rank value”, right?
- In line 268, “n” is associated with the shape of V and aslo to the  number of merged tasks, is this notation correct?

**Questions:**

1. Table 1 vs. Table 4: The results for your method differ between Tables 1 and 4. Could you clarify why this is the case?
2. In table 4 Subspace Boosting 4 uses LiNeS while other baselines do not, why, is the comparison fair?
3. Table 10,  Random selection: In Table 10, you average over 20 random selections when merging 8 out of 20 models. Why not also report the standard deviation to show the variance of random selection?
4.  In Figure 3, the largest singular values seem to scale roughly linearly with the number of merged experts (e.g., almost 0.05 for 4 experts,  close to 0.10 for 8, and almost 0.20 for 20).
This appears consistent with what one would expect if the overall Frobenius norm of the merged matrix increases linearly with the number of merged task vectors, meaning the curves could differ mainly by a global scaling factor rather than by a change in shape.
Did you normalize the merged weight matrices (e.g., by dividing by the Frobenius norm or the largest singular value) before plotting the singular-value distributions in Fig. 3?
If not, could the apparent steepening of the spectra be explained by such scaling rather than by true “rank collapse”?
5. $\beta$ is tuned over the set {0, 0.01, 0.02}. How this range was chosen, and whether a broader search might affect results?
6. You mention that Subspace Boosting is faster than competing methods. Could you explain which component (e.g., decomposition, projection step, or optimization) contributes most to this improvement?
7. Have you considered including the Singular Task Interference metric (Gagiurlo et al.) as an additional as a way to interpret task similarity?

---

> ### Author Response · Authors · 2025-11-23
> **Reply Reviewer AddN part 1**
>
> We deeply thank the reviewer for their in-depth review and their constructive feedback regarding our work.
>
>  In particular, we would like to highlight that **we have added an extensive new theoretical section 3.1 in the paper**, proving that _rank collapse is inevitable for Task Arithmetic methods_. We believe the additional propositions that we have included in our new revision **significantly enhance the overall paper and formal notion of “rank collapse”**.
>
> We address your concerns in detail below.
>
> ---
>
> **Theoretical formulation of Rank Collapse**
>
> We thank the reviewer for highlighting this limitation of our previous revision. **Following similar constructive feedback from the other reviewers, we have now added a completely new theoretical section, formalizing rank collapse**. In addition to the empirical evidence of rank collapse in our previous revision, **we now have also theoretically proved that rank collapse is inherent to any Task Arithmetic-based method (Propositions 1-3. - main).** We also provide theoretically grounded justification for Subspace Boosting in Proposition 4 (supp).
>
> ---
>
> **Subspace Boosting SVD-step**
>
> We thank the reviewer for the comment. We aim to provide pseudocode that closely resembles the functional code, provided in the supplementary material. Equivalent to the real code, we apply SVD inside the Subspace Boosting algorithm and do not provide it as an argument. Please let us know if we could clarify your question better.
>
> ---
>
> **Typos**
>
> We have fixed these minor typos in our previous revision and now use uppercase “N” to denote the number of tasks, preventing further confusion.
>
> ---
>
> **Differences of results reported in Tab1 and Tab4**
>
> Table 1 evaluates the method using the standard number of merging coefficient interpolation points as the baseline methods, i.e., from 0.0 to 1.0 in increments of 0.1. Whereas Table 4 shows the results for TSV-M and Iso-C/CTS, which use a larger number of interpolation points, i.e., from 0.0, to 3.0. Therefore, we evaluate our best-performing method with the same number of interpolation points as the TSV-M and Iso-C/CTS to enable fair comparison. We explicitly state this in our subsection **State-of-the-Art Comparison**.
>
> ---
>
> **Reason for using LiNeS in SOTA comparison**
>
> Our intention with Table 4 is to show that solving rank collapse for simple methods drastically improves their performance, reaching state-of-the-art performance. The other baselines in Table 4 are significantly more complex than our simple Task Arithmetic approach, as also seen by the significant computational efficiency of our approach. These results are meant to highlight the unexpected and counterintuitive improvement by simply resolving rank collapse of very naive merging methods.
>
>  ---
>
> **Standard Deviation for random expert selection**
>
>
> We thank the reviewer for making us aware of this and we have now added the standard deviations in the Table 10. It is noticeable that the random selection yields a mean accuracy of 67.5 with the standard deviation of 2.3, while using our HO-GSVD selection strategy the accuracy is 69.7. These results show that HO-GSVD reaches higher performance, far from the average performance of random selection.
>
> ---
>
> **Linear scaling of the maximum singular value when number of task increases**
>
> This is an extremely astute observation by the reviewer that prompted us to formally understand the reason for this linear scaling. We investigated this further and formally proved the reason for this linear scaling and to further understand the reasoning behind the “steepening of the spectra”, as mentioned by the reviewer.
>
> Following this investigation, we prove this linear scaling in Proposition 3 and show that this linear scaling does not affect our rank collapse hypothesis.
>
> In essence, singular values corresponding to common components, such as the largest singular values, scale linearly $O(N)$ with the number of merged models N. Whereas, task-specific singular values scale $O(\sqrt{N})$. Since the top singular values correspond to common components, we see the linear scaling that was observed by the reviewer in Fig.3. Additionally, we empirically illustrate (in Appendix E4) that the unique components scale in $O(\sqrt{N})$, while the common ones scale in  $O(N)$.
>
>
> However, this does not affect our rank collapse conclusion. Since $O(N)/O(\sqrt{N}) = O(\sqrt{N})$, as more models are merged, i.e., $N&rarr; \infty$, the common components will dominate the task-specific information, leading to the mass of the energy to be placed on the dimensions corresponding to the common components. In other words, as more models are merged, the relative importance of the task-specific information actually diminishes.

---

> ### Author Response · Authors · 2025-11-23
> **Reply Reviewer AddN part 2**
>
> ---
>
> **Robustness of $\beta$ hyper-parameter in Subspace Boosting**
>
> The range for $\beta$  was chosen based on extensive empirical evaluation. Initially, we evaluated using the range from 0.0 to 0.1. In increments of 0.01 and created ablations depending on the different weight types with the base Vision transformer model. From our extensive experiments, we concluded that the range {0.0, 0.1, 0.2} is a good balance of finding the optimal performance while at the same time minimizing extensive hyperparameter evaluation. The robustness of the hyperparameter and the hyperparameter search can be seen in Tables 3 and 7, where the maximum observed difference performance is only 0.7% for 20 tasks with ViT-B/16  when varying $\beta$.
>
> As is the case with any hyperparameter, a more extensive search could further improve performance at the cost of expensive hyperparameter evaluation. However, we instead focus on computational efficiency and simplicity and less on hyperparameter tuning. In fact, we advise the reader to use $\beta=0.0$ by default for strong performance without the need for additional hyperparameter tuning. The results in Table 2 for the Language Results actually involve no hyperparameter tuning and we simply utilize our previous recommendation of setting  $\beta=0.0$, while simultaneously exhibiting strong performance (improving the TA+LiNeS results from 67.6 -> 75.3 obtained by Subspace Boosting with default  $\beta=0.0$).
>
> ---
>
> **Subspace Boosting running time**
>
> Our method is significantly more efficient than competing methods for the following reasons:
> TSV-M performs SVD on every matrix individually to get Task Singular Vectors. In essence, this is O(N), where N is the number of tasks for merging and afterwards once more for the decorrelation (O(1)). On the other hand, Subspace Boosting only applies one SVD on the merged matrix, regardless of the number of tasks, which is O(1). Hence, TSV-M scales linearly whereas Subspace Boosting scales in O(1), making our method significantly more efficient.
>
> Since Iso-CTS is an extension on top of TSV-M, it is even more computationally heavy than TSV-M and therefore, our method. The benefit of our method irrespective of the number of merged models would be especially evident for larger models. The running time for our Subspace Boosting and TSV-M is presented in Table 6, showing that Subspace Boosting is 6x faster than TSV-M.
>
> ---
>
> **Singular Task Interference metric for interpretability**
>
> While the Singular Task Interference metric from Gargiulo et al.  is a valid approach for evaluating interference between tasks, we ultimately argue that the metric is naive and not granular enough. Furthermore, we argue that the Alignment Matrix provides a superior method for interpreting task similarity.
>
> Firstly, the Singular Task Interference (STI) only evaluates the orthogonality of the task vectors. By computing the inner products of the U.TU and V.TV which contain the concatenated task singular vectors, it computes how orthogonal or overlapping the different directions are. However, it still decomposes the task vectors using individual U_i and V_i matrices. Afterwards, it aggregates it into one scalar result, utilizing the L1 norm. However, the authors do not argue why this is an optimal way to evaluate the interference and it seems to be more of a heuristic than a mathematically grounded manner of evaluating task similarity. Therefore, this is just an approximate metric of evaluating task similarity or interference.
>
> On the other hand, we provide a more mathematically rigorous way to evaluate task similarity using the Alignment matrix. By using HO-GSVD, we utilize one shared basis V across all tasks. This allows for direct comparison of how different tasks are utilizing the same dimension. This type of granularity cannot be achieved with the Singular Task Interference metric. This also allows one to evaluate not only how much models interfere with each other, but how they share the underlying parameter space (common vs unique components).
>
> Moreover, unlike the Singular Task Interference metric, our approach allows us to explicitly disentangle the signal by categorizing “common” and “unique” subspaces. This provides us the ability to distinguish whether a dimension is crucial for multiple tasks (high interference) or unique to one (high alignment), providing a better interpretation of task similarity.
>
> Finally, unlike the STI by the original authors, we show that our Alignment matrix effectively evaluates the interference by successfully selecting a subset of models that merge well, outperforming random baselines in Tables 5 and 10.

---

### Author Response · Authors · 2025-11-23
**Summary of Changes**

We deeply appreciate the reviewers' feedback that improved our work significantly.  The reviewers agreed that our work is _“relevant to the community”_ (AddN), and that we identified _“a critical limitation in existing model merging approaches”_ (NH2N), meaning Rank Collapse, _“explaining it from a task vector space perspective”_ (Eqcf). Reviewers also noticed that our method Subspace Boosting is _“highly practical”_ (Eqcf),  _“more efficient”_ (RgdQ), _“applicable to any existing task-arithmetic variant”_ (RgdQ)  and _“computationally efficient”_ (AddN).


We appreciate the reviewers for encouraging us to strengthen the theoretical foundations of this work, especially regarding rank collapse and the effect of our Subspace boosting. Our revised manuscript includes Propositions 1-3, which **theoretically demonstrate that rank collapse is inherent to any Task Arithmetic-based method**.  In other words, we prove that Task Arithmetic model merging inevitably achieves lower performance as more models are merged.  Additionally, Proposition 4 provides a theoretically grounded justification for our proposed Subspace Boosting technique.
In conclusion, **we now  provide the first formal proof in the literature diagnosing Rank Collapse as an inevitable consequence of orthogonality in high-dimensional task vectors**.

We hope we have addressed every point raised by the reviewers. If there are any remaining questions or areas that require additional detail, please let us know.

---

### Author Response · Authors · 2025-12-03
**Summary of changes based on reviewers' concerns**

### We highlight the specific actions taken to address the reviewers’ concerns below:

-----

**“Rank collapse requires more formal explanation”** (Reviewer AddN, RgdQ):

We added a completely new theoretical section, formalizing rank collapse. In addition to the empirical evidence of rank collapse in our previous revision, we theoretically proved that rank collapse is inherent to any Task Arithmetic-based method (Propositions 1, 2, 3 in the main paper). Additionally, we empirically illustrated (Appendix E4) that the unique components scale in $O(\sqrt{N})$, while the common ones scale in  $O(N)$.

----
**“Potential error amplification of Subspace boosting”** (Reviewer NH2N)

We added Proposition 4 to the Appendix, which establishes the Signal-to-noise tradeoff of Subspace Boosting. We proved that under mild assumptions, the task-specific information recovered by Subspace Boosting exceeds the error introduced by boosting the noise.

---

**“Hyperparameter sensitivity”** (Reviewer NH2N, Eqcf)

We demonstrated that the hyperparameter $\beta$ of Subspace boosting is robust to tuning in Table 3 (main). Additionally, we noted in the paper that users can simply choose  to omit this hyperparameter tuning (using the default $\beta=0$). In fact, for the NLP results in Table 2 (main), we did not perform any hyperparameter tuning and used the default setting. This default configuration still improved the TA+LiNeS results significantly (67.6 $\to$ 75.3).

---

**“Stable rank reflects the impact of rank Collapse”** (Reviewer  Eqcf)

Following Proposition 1, we proved in Proposition 2 that the stable rank converges to the stable rank of the common component as more models are averaged. Therefore, this demonstrates that the stable rank is an accurate and theoretical grounded metric for evaluating rank collapse.

---
**“Improvement on experiments on different architectures/modality types”** (Reviewer  Eqcf)

We mathematically proved that rank collapse is inevitable (Propositions 1-3) and inherent to any Task Arithmetic-based method. Therefore, our Subspace boosting method that alleviates rank collapse improves the performance of any TA-based method (as proved in Proposition 4 in the Appendix). Moreover, we performed experiments on different domains (vision Tab1, NLP Tab2) and different architectures (ViT Tab1, T5 Tab2).

---

**“Limitied Novelty and SotA results compared to Iso-c/Iso-CTS”** (Reviewer RgdQ)

Unlike Iso-c/Iso-CTS [1], we are the first to discover, quantify and prove that rank collapse is inevitable for Task Arithmetic-based methods. We also propose the first interpretable model merging framework based on HO-GSVD. While the primary  goal of our work is to investigate _why_ merging performance declines as more models are merged, our method remains highly competitive,  outperforming Iso-CTS on 4/9 tasks (Tab4).

[1] "No Task Left Behind: Isotropic Model Merging with Common and Task-Specific Subspaces." Marczak, et al. ICML 2025.

---

### Meta-Review · Area_Chair_F6PX · 2026-01-06

**Summary:**

The paper provides both empirical and theoretical evidence that as more models are merged in model merging, the overall performance degrades. The work claims that this is due to a phenomenon called "rank collapse", where common information in all tasks dominates over task specific information. A method to mitigate rank collapse is proposed, which boosts smaller singular values of the task vector matrices and increases them up to a threshold if they fall below a certain threshold.  The method is shown to improve basic model merging methods, and performs comparably to recent state-of-the art methods.

From the initial reviews, the overall consensus among reviewers pointed towards rejecting this work. While some concerns were addressed in the rebuttal, a major concern raised by almost all reviewers about the validity of the rank collapse phenomenon remains. The rebuttal added a new detailed theoretical analysis of the collapse phenomenon and uses that analysis to further support the claims. However, it is unclear whether reviewers would be convinced by that and looking at the results I am unsure myself. In general, given their importance, the added theoretical results will require a rigorous and detailed check for soundness in light of the strong claims.

Given that the method performs comparably to other state-of-the-art model merging methods, introduces an additional hyperparameter and reviewers all had doubts about the rank collapse phenomenon, the paper is not recommended for acceptance in its current form.

**Reviewer Concerns:**

> The rank collapse phenomenon could benefit from a formal explanation (AddN), doubts about validity of rank collapse phenomenon (Eqcf, RgdQ)
- A new theoretical section has been added to the paper, but such larger change needs careful checking for soundness. The concerns are still outstanding.

> Extra hyperparameter \beta is required (NH2N, Eqcf, RgdQ)
- The rebuttal mostly argued that setting it to zero can give decent results, but the resulting method can fall behind other SVD based methods which do not use any hyper-parameters.

> Error amplification through boosting (NH2N)
- Seems to be addressed, an analysis under signal to noise ratio assumptions has been added to appendix.

> Novelty limited over previous works of Marczak et al, Gargiulo et al (RgdQ)
- The rebuttal emphasized the fact that the work is the first to diagnose and prove that rank collapse happens and performance degrades as more models are merged, the concern seems mostly addressed.

> Interpretable merging oversold (RgdQ)
- Partially addressed, rebuttal argues that interpretability is secondary to the work, but that some insights can be gained from alignment matrix.

**Reviewer Scores:**

Here are my estimates how the reviewers would have changed their score:

AddN: 4 -> 6
NH2N:  2 -> 2
Eqcf: 4 -> 6
RgdQ: 4 -> 4

---

### Decision · Program_Chairs · 2026-01-26

Reject